



# Multi-year evaluation of airborne geodetic surveys to estimate seasonal mass balance, Columbia and Rocky Mountains, Canada

**Ben M. Pelto**[1], **Brian Menounos**[1], and **Shawn J. Marshall**[2]

[1]Natural Resources and Environmental Studies Institute and Geography Program,
University of Northern British Columbia, Prince George, V2N 4Z9, Canada
[2]Department of Geography, University of Calgary, Calgary, T2N 1N4, Canada

**Correspondence:** Ben M. Pelto (pelto@unbc.ca)

**Abstract.** CE1 Seasonal measurements of glacier mass balance provide insight into the relation between climate forcing and glacier change. To evaluate the feasibility of using remotely sensed methods to assess seasonal balance, we completed tandem airborne laser scanning (ALS) surveys and field-based glaciological measurements over a 4-year period for six alpine glaciers that lie in the Columbia and Rocky Mountains CE2, near the headwaters of the Columbia River, British Columbia, Canada. We calculated annual geodetic balance using coregistered late summer digital elevation models (DEMs) and distributed estimates of density based on surface classification of ice, snow, and firn surfaces. Winter balance was derived using coregistered late summer and spring DEMs, as well as density measurements from regional snow survey observations and our glaciological measurements. Geodetic summer balance was calculated as the difference between winter and annual balance. Winter mass balance from our glaciological observations averaged $1.95 \pm 0.09$ m w.e. (meter water equivalent), 4 % larger than those derived from geodetic surveys. Average glaciological summer and annual balance were 3 % smaller and 3 % larger, respectively, than our geodetic estimates. We find that distributing snow, firn, and ice density based on surface classification has a greater influence on geodetic annual mass change than the density values themselves. Our results demonstrate that accurate assessments of seasonal mass change can be produced using ALS over a series of glaciers spanning several mountain ranges. Such agreement over multiple seasons, years, and glaciers demonstrates the ability of high-resolution geodetic methods to increase the number of glaciers where seasonal mass balance can be reliably estimated.

## 1 Introduction

Glaciers are in rapid retreat across western Canada (Menounos et al., 2019). Deglaciation is projected to have pronounced impacts on streamflow in western Canada (Clarke et al., 2015), with the greatest reductions in August and September streamflow as glaciers shrink (Huss and Hock, 2018; Jost et al., 2012). In the Canadian Columbia River basin, peak glacier runoff from ice wastage is either currently underway (Huss and Hock, 2018) or will occur within the next decade (Clarke et al., 2015). Improved projections of changes in glacier runoff will require refined treatment of seasonally varying processes that nourish and deplete glaciers, namely the redistribution of snow by wind and gravitational processes and changes in surface albedo. Seasonal mass balance records are also required to calibrate and validate these physically based mass balance models. These records do not currently exist for the Columbia River basin, however.

In addition to their use in refining estimates of future changes in glacier runoff, mass balance observations provide a valuable synopsis of a glacier's mass budget and its implications for glacier runoff (Jost et al., 2012; Ragettli et al., 2016; Stahl and Moore, 2006), water storage, regional climate (Huss et al., 2008; Radić and Hock, 2014), and contribution to sea level rise (Huss and Hock, 2018). Glacier mass balance is a function of accumulation and ablation processes,

responding directly to meteorological forcing at timescales of a season or more (Oerlemans et al., 1998). Measurement of seasonal mass change via in situ and geodetic methods provides a means to assess the importance of meteorological drivers of glacier nourishment and melt. These observations can reveal trends and patterns in glacier mass evolution and are valuable calibration and validation datasets for global (Huss and Hock, 2018; Maussion et al., 2019) and regional glacier models (Clarke et al., 2015), as well as for ingestion into regional hydrologic models (Schnorbus et al., 2014).

Seasonal balance is logistically and financially difficult and globally few seasonal mass balance records exist (Ohmura, 2011). Currently, seasonal balance measurements for western Canadian glaciers are not publicly available (WGMS, 2018). Seasonal snowpack forms a critical component of glacier mass balance (Østrem and Brugman, 1991); it controls the volume and timing of runoff in the snowmelt-dominated tributaries to the Columbia River (Brahney et al., 2017). Like many regions (Barnett et al., 2005), high-elevation snow and precipitation records are limited in the Columbia River basin of Canada. Snow data are routinely only monitored at or below treeline, and much of the basin, including its glaciated terrain, exists above this elevation. Some models suggest snowpack may be increasing at high elevations (Schnorbus et al., 2014), though existing snow observations below treeline indicate decreased water equivalent through the 1980–2011 period (Brahney et al., 2017). This data gap hinders accurate estimates of alpine snowpack in the region, which is critical for glacier nourishment, ecosystems, hydropower, and flood forecasts (Hamlet et al., 2005).

Geodetic methods are now regularly used to measure seasonal snow depth on glaciers via surface (Helfricht et al., 2014; McGrath et al., 2015) or helicopter-borne ground-penetrating radar (Dadic et al., 2010; Machguth et al., 2006; Sold et al., 2015), airborne laser scanning (ALS) surveys (Helfricht et al., 2012, 2014; Sold et al., 2013), airborne photogrammetry (Nolan et al., 2015), and stereoscopic satellite imagery (Belart et al., 2017). Geodetic surveys offer the ability to greatly expand the number of glaciers over which snow depth and mass change estimates can be made (Berthier et al., 2014; Nolan et al., 2015). For hydrological applications, snow depth must be converted into snow water equivalent (SWE), and thus snow density must be known or estimated. Physical modeling of snow density is difficult (Sold et al., 2015), and in situ density measurements are sparse and are expensive in terms of cost and effort. Density measurements for snow over glacier surfaces often show little relation to either elevation or snow depth (Fausto et al., 2018; Machguth et al., 2006; McGrath et al., 2015). Density thus introduces uncertainty to geodetic winter SWE estimates which are vital to calibrate and validate hydrological modeling and to measure seasonal mass balance (Belart et al., 2017; Sold et al., 2013). The primary objective of our study is to evaluate the reliability of geodetic surveys and density assumptions for estimation of seasonal mass change of temperate glaciers over multiple years.

## 1.1 Study area

### 1.1.1 Columbia Mountains

The transboundary Columbia River basin (668 000 km$^2$) spans seven US states and the province of British Columbia (BC), Canada. The Canadian portion of the basin represents 15 % of the watershed's total area yet provides between 30 %–40 % of its total runoff, largely due to the presence of mountainous terrain with high amounts of orographic precipitation and extensive glacial cover (Cohen et al., 2000; Hamlet and Lettenmaier, 1999). There are 2200 glaciers covering 1760 km$^2$ in the Columbia Mountains (Bolch et al., 2010); these glaciers primarily exist within the Cariboo, Monashee, Selkirk, and Purcell ranges, with the highest elevations rising to over 3000 m above sea level (a.s.l.).

The Columbia Mountains are transitional between maritime and continental (Demarchi, 2011). Monthly average temperatures in the Canadian Columbia River basin (elevation range from 420 to 3700 m a.s.l.) range from −9.2 °C in January to +13.3 °C in July (Najafi et al., 2017; Schnorbus et al., 2014). General circulation is dominated by westerly flow, which brings consistent Pacific moisture, particularly in the winter months. Approximately 65 % of annual precipitation falls as snow, with snowfall possible throughout the year (Schnorbus et al., 2014). The snow accumulation season in both the Columbia and Canadian Rocky Mountains extends from October to May. The summer melt season runs from May through September. From 1981 to 2010, Rogers Pass, located in the center of the Columbia Mountains (Fig. 1), at an elevation of 1330 m a.s.l., had an average annual temperature of +1.9 °C, an average winter (December–February) temperature of −8.0 °C, and experienced 1056 ± 49 mm w.e. (millimeter water equivalent) of precipitation through the accumulation season (October–April) (Environment Canada, 2019).

### 1.1.2 Rocky Mountains

The southern Canadian Rocky Mountains are located east of the Columbia Mountains (Fig. 1) across the Rocky Mountain Trench and are home to 1090 glaciers covering 1350 km$^2$ (Bolch et al., 2010).

The eastern slopes of the Canadian Rocky Mountains experience a continental climate with mild summers and cold winters. However, winter precipitation along the continental divide is greatly influenced by moist Pacific air masses, with persistent westerly flow driving orographic uplift on the western flanks of the Rocky Mountains (Sinclair and Marshall, 2009). This combination of continental and maritime influences fosters extensive glaciation along the continental divide in the Canadian Rockies, with glaciers at el-

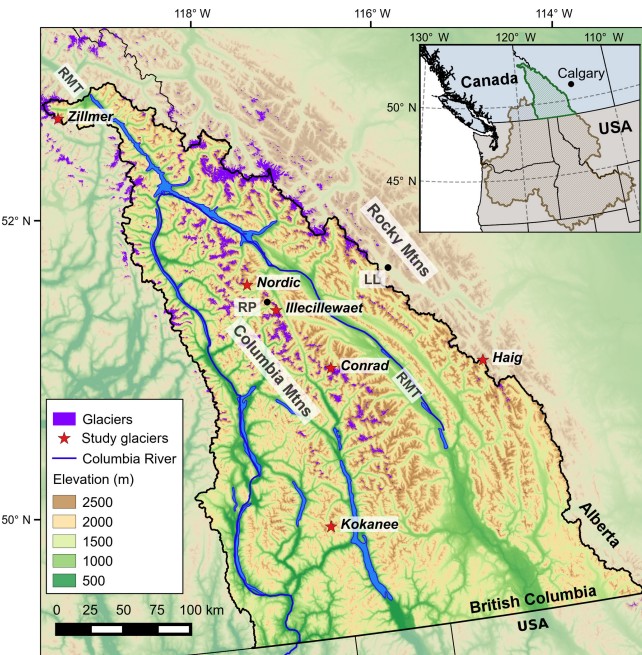

**Figure 1.** Map of the Canadian Columbia River basin (black outline, brighter topography) and locations of study sites. Inset shows regional context of the Canadian portion of the Columbia River basin which contributes to the river where it crosses the international border (green). The remainder of the basin is also depicted (brown). The Columbia and Rocky Mountains are separated by the Rocky Mountain Trench (RMT). Weather stations (black dots) at Rogers Pass (RP) and Lake Louise (LL) are referred to in the introduction. Map coordinates are in NAD83/BCAlbers.

evations from 2200 to 3500 m a.s.l. on the eastern slopes. From 1981 to 2010, Lake Louise, located in the center of the southern Canadian Rockies (Fig. 1), at an elevation of 1524 m a.s.l., had an average annual temperature of +0.2 °C, an average winter temperature of −11.6 °C, and experienced 298 ± 9 mm w.e. of precipitation through the accumulation season (Environment Canada, 2019). As evidenced by comparing Lake Louise and Rogers Pass, the Canadian Rocky Mountains are drier and colder in winter than the Columbia Mountains.

## 2 Data and methods

### 2.1 Study sites

Over the period 2014–2018 we measured seasonal mass balance of six alpine glaciers (Table 1): (1) Zillmer Glacier (5.4 km$^2$) in the Cariboo Mountains; (2) Nordic Glacier (3.4 km$^2$) and (3) Illecillewaet Glacier (7.7 km$^2$) in the Selkirk Mountains; (4) Conrad Glacier (11.5 km$^2$) and (4) Kokanee Glacier (1.8 km$^2$) in the Purcell Mountains; and (5) Haig Glacier (2.6 km$^2$), which straddles the continental

divide. Haig Glacier is in the Rocky Mountains, whereas the other five glaciers lie in the Columbia Mountains.

### 2.2 Geodetic mass balance

We performed repeat fixed-wing ALS surveys from late summer 2014 to late summer 2016 (Table 2) using a Riegl VQ-580 infrared (1024 micron) laser scanner with dedicated Applanix POS AV Global Navigation Satellite System (GNSS) inertial measurement unit (IMU). Later surveys used the same GNSS IMU and a Riegl Q-780 infrared (1024 micron) laser scanner. The VQ-580 and Q-780 were respectively flown at flying heights of around 500 and 2500 m above the terrain that yielded swath widths of 500–1000 and 2000–3000 m. Effective sampling diameter was 10–20 cm per laser shot. We planned the airborne surveys with 53 % overlap between flight lines to yield return point densities that averaged 1–3 laser shots m$^{-2}$ (Table 2) and to minimize systematic bias from off-nadir laser shots.

#### 2.2.1 ALS post-processing

Post-processing of the ALS survey flight trajectory data used the PosPac Mobile Mapping Suite (Applanix), with Trimble CenterPoint RTX with vertical and horizontal positional uncertainties that were typically better than ±15 cm (1σ). We post-processed point clouds and exported data into LAS (lidar data exchange file) files, a binary file format that can be efficiently processed with LAStools (https://rapidlasso.com/lastools/, last access: 30 May 2017, Isenburg, 2014). We used the LAStools las2dem algorithm to create 1 m resolution digital elevation models (DEMs). Las2dem triangulates ground classified ALS points from LAS files into a temporary triangulated irregular network (TIN). A DEM is then created from this using nearest-neighbor interpolation. Given an average point density of greater than 2 points m$^{-2}$ (Table 2), little interpolation was required. We coregistered all DEMs following the method detailed in Nuth and Kääb (2011). For late summer surveys, one master DEM was chosen and all other late summer DEMs were coregistered to that DEM for stable terrain (e.g., off-glacier) only. Stable terrain was identified in satellite imagery and excluded forests, lakes, and ice- and snow-covered areas. For winter DEMs, the previous late summer DEM was used as the master DEM to mitigate against any surface height changes in areas defined as stable terrain, due to processes such as rockfall or vegetation height change. During the spring surveys, there was little to no snow-free terrain, except rocky features with extreme slopes which are not used in the coregistration (slope > 40° excluded). We thus did not apply any vertical shift during coregistration of winter DEMs.

We utilized satellite imagery from Landsat 7 and 8, Sentinel-2, and PlanetScope at 30, 10, and 3–5 m resolution respectively (Bevington et al., 2018), to guide surface classification used to coregister DEMs and calculate geodetic

**Table 1.** Glacier-specific details. Firn ratio refers to the area of a glacier covered by multi-year firn, which is the combination of accumulation area and exposed firn from 2015 imagery.

| Glacier | Area (km$^2$) | Max elev. (m) | Min elev. (m) | Range (m) | Mean elev. (m) | Length (km) | Firn ratio | Aspect |
|---|---|---|---|---|---|---|---|---|
| Zillmer | 5.43 | 2860 | 1860 | 1000 | 2380 | 5.59 | 0.59 | NW |
| Nordic | 3.39 | 2990 | 2065 | 925 | 2515 | 3.30 | 0.62 | N |
| Illecillewaet | 7.72 | 2908 | 2147 | 761 | 2532 | 4.29 | 0.48 | WNW |
| Haig | 2.62 | 2870 | 2461 | 409 | 2660 | 2.45 | 0.06 | SE |
| Conrad | 11.45 | 3235 | 1825 | 1410 | 2595 | 12.18 | 0.58 | N |
| Kokanee | 1.79 | 2805 | 2220 | 585 | 2585 | 2.20 | 0.48 | N |

**Table 2.** Date and number of observation locations ($n$) for glaciological visits and geodetic acquisition dates and point density. Field dates are the median date of glacier visit.

| Year | Glacier | Autumn glac. (m/d/y) | $n$ | Autumn ALS (m/d/y) | Cover (%) | Points m$^{-2}$ | Winter glac. (m/d/y) | $n$ | Winter ALS (m/d/y) | Cover (%) | Points m$^{-2}$ |
|---|---|---|---|---|---|---|---|---|---|---|---|
| 2015 | Zillmer | 8/23/2015 | 23 | 10/3/2015 | 100 | 2.75 | 5/30/2015 | 20 | 4/19/2015 | 100 | 3.34 |
| 2016 | Zillmer | 8/15/2016 | 23 | 9/14/2016 | 100 | 2.44 | 4/14/2016 | 46 | 4/18/2016 | 100 | 3.69 |
| 2017 | Zillmer | 8/22/2017 | 26 | 11/3/2017 | 100 | 1.49 | 4/13/2017 | 31 | 5/20/2017 | 100 | 0.80 |
| 2018 | Zillmer | – | – | – | – | – | 5/19/2018 | 42 | 4/29/2018 | 100 | 4.37 |
| 2014 | Nordic | 8/29/2014 | 8 | 9/11/2014 | 100 | 8.71 | 4/27/2014 | 16 | – | – | – |
| 2015 | Nordic | 8/31/2015 | 11 | 9/11/2015 | 99 | 1.99 | 5/1/2015 | 20 | 4/19/2015 | 100 | 3.04 |
| 2016 | Nordic | 8/21/2016 | 21 | 9/12/2016 | 99 | 3.27 | 5/2/2016 | 28 | 4/17/2016 | 100 | 3.21 |
| 2017 | Nordic | 9/14/2017 | 18 | 9/27/2017 | 100 | 2.35 | 5/1/2017 | 21 | 5/21/2017 | 100 | 0.96 |
| 2018 | Nordic | – | – | – | – | – | 5/1/2018 | 21 | 4/26/2018 | 100 | 1.84 |
| 2015 | Illecillewaet | 9/24/2015 | 9 | 9/11/2015 | 97 | 1.02 | – | – | 4/19/2015 | 100 | 2.31 |
| 2016 | Illecillewaet | 9/13/2016 | 7 | 9/12/2016 | 100 | 1.37 | – | – | 4/17/2016 | 100 | 2.50 |
| 2017 | Illecillewaet | 9/27/2017 | 7 | 9/17/2017 | 100 | 2.59 | 5/19/2017 | 3 | 5/21/2017 | 100 | 1.22 |
| 2018 | Illecillewaet | – | – | – | – | – | – | – | 4/26/2018 | 100 | 1.64 |
| 2015 | Haig | 9/12/2015 | 2 | 9/12/2015 | 100 | 0.93 | 5/12/2015 | 33 | 4/20/2015 | 100 | 2.89 |
| 2016 | Haig | 9/13/2016 | 1 | 9/13/2016 | 100 | 1.85 | 5/18/2016 | 33 | 4/17/2016 | 100 | 2.64 |
| 2017 | Haig | 9/16/2017 | 1 | 9/16/2017 | 97 | 4.82 | 5/12/2017 | 33 | 5/21/2017 | 100 | 1.09 |
| 2018 | Haig | – | – | – | – | – | – | – | 4/27/2018 | 100 | 3.23 |
| 2014 | Conrad | 9/4/2014 | 7 | 9/11/2014 | 100 | 10.38 | – | – | – | – | – |
| 2015 | Conrad | 9/5/2015 | 9 | 9/12/2015 | 92 | 1.35 | 4/23/2015 | 38 | 4/20/2015 | 100 | 3.58 |
| 2016 | Conrad | 8/28/2016 | 31 | 9/12/2016 | 100 | 2.45 | 4/26/2016 | 44 | 4/17/2016 | 100 | 2.45 |
| 2017 | Conrad | 9/10/2017 | 42 | 9/17/2017 | 94 | 3.70 | 5/15/2017 | 59 | 5/21/2017 | 100 | 1.29 |
| 2018 | Conrad | – | – | – | – | – | 4/24/2018 | 56 | 4/26/2018 | 100 | 1.84 |
| 2015 | Kokanee | 8/27/2015 | 11 | 9/12/2015 | 100 | 1.04 | 4/20/2015 | 20 | 4/19/2015 | 100 | 2.99 |
| 2016 | Kokanee | 9/5/2016 | 23 | 9/13/2016 | 100 | 2.07 | 4/19/2016 | 33 | 4/17/2016 | 100 | 2.77 |
| 2017 | Kokanee | 9/19/2017 | 15 | 9/16/2017 | 83 | 2.63 | 4/17/2017 | 23 | 5/21/2017 | 100 | 0.92 |
| 2018 | Kokanee | – | – | – | – | – | 4/18/2018 | 21 | 4/26/2018 | 100 | 1.33 |

mass change. We selected the latest snow-free imagery from September or late August and used a band ratio and threshold method (Kääb, 2005) to classify areas of snow, firn, and ice. In some cases, we manually corrected surface maps where our automated methods failed to differentiate between firn and snow surfaces.

To calculate annual mass change ($B_a$), we (1) difference two DEMs to create a height change DEM ($\Delta$DEM); (2) bias correct the height change by the mean height difference over stable terrain between two DEMs after coregistration (Bias$_{\Delta h}$, Table 3); (3) derive a mask based on surface classi-

fication of ice, firn, and snow from satellite imagery (Fig. S1 in the Supplement); and then (4) apply the density of each respective surface type (Table 4), to the $\Delta$DEM to calculate mass balance. We chose not to use digital terrain models (DTMs), which represent gridded elevation based on last returns from the laser scanner, since our gridding algorithms employed in LAStools filled crevasses and did not preserve sharp ridges that aided in coregistration of the DEMs.

Annual glacier mass balance is defined as the sum of accumulation and ablation throughout the balance year (Cuffey and Paterson, 2010), which can be expressed as the sum of

**Table 3.** Seasonal balance and uncertainty estimates for geodetic (geod) and glaciological mass balance (glac) in meter water equivalent (m w.e.). Kinematic-GPS-survey-derived corrections applied to glaciological data (surv.corr). Statistical analysis of the DEMs over stable terrain include NMAD, median height difference, and bias correction applied over the glacier (Bias$_{\Delta h}$). Mean density of $B_{a\_geod}$ is $\overline{\rho}$. Average values include only cases where both geodetic and glaciologic data were collected. $B_{w\_geod.gl}$ is calculated using glaciological densities (Table S1), and $B_{w\_geod.ss}$ is calculated using snow survey data (Fig. 2). Listed $B_{s\_geod}$ is derived using $B_{w\_geod.ss}$. Regional late summer snow density (Table 5) was used to calculate $B_{a\_geod}$.

| Year | Glacier | $B_{w\_geod.gl} \pm \sigma_{geod.bw}$ | $B_{w\_geod.sc} \pm \sigma_{geod.bw}$ | $B_{s\_geod} \pm \sigma_{geod.bs}$ | $B_{a\_geod} \pm \sigma_{geod.ba}$ | $B_{w\_glac} \pm \sigma_{glac.bw}$ | $B_{s\_glac} \pm \sigma_{glac.bs}$ | $B_{a\_glac} \pm \sigma_{glac.ba}$ | $B_{w\_surv.corr}$ | $B_{a\_surv.corr}$ | AAR | ELA (m) | NMAD $B_a$ (m) | NMAD $B_w$ (m) | Median $B_{a\Delta h}$ (m) | Bias$_{\Delta h}$ (m) | $\overline{\rho}$ (kg m$^{-3}$) |
|---|---|---|---|---|---|---|---|---|---|---|---|---|---|---|---|---|---|
| 2018 | Zillmer | 1.70 ± 0.19 | 1.75 ± 0.20 | – | – | 1.65 ± 0.17 | – | – | −0.15 | – | – | – | – | 1.4 | – | – | – |
| 2018 | Nordic | 1.87 ± 0.26 | 2.07 ± 0.27 | – | – | 2.18 ± 0.14 | – | – | −0.04 | – | – | – | – | 1.76 | – | – | – |
| 2018 | Illecillewaet | 1.61 ± 0.17 | 1.65 ± 0.18 | – | – | – | – | – | – | – | – | – | – | 2.26 | – | – | – |
| 2018 | Haig | 1.25 ± 0.15 | 1.31 ± 0.19 | – | – | 1.42 ± 0.15 | – | – | – | – | – | – | – | 1.83 | – | – | – |
| 2018 | Conrad | 1.62 ± 0.21 | 1.84 ± 0.23 | – | – | 1.83 ± 0.12 | – | – | 0.00 | – | – | – | – | 2.34 | – | – | – |
| 2018 | Kokanee | 2.07 ± 0.25 | 2.31 ± 0.26 | – | – | 2.25 ± 0.13 | – | – | 0.01 | – | – | – | – | 1.76 | – | – | – |
| 2017 | Zillmer | 2.12 ± 0.24 | 2.03 ± 0.25 | −2.70 ± 0.27 | −0.67 ± 0.10 | 1.93 ± 0.26 | −2.44 ± 0.35 | −0.51 ± 0.23 | 0.15 | −0.31 | 0.48 | 2440 | 0.6 | 1.83 | −0.1 | −0.05 | 729 ± 45 |
| 2017 | Nordic | 2.14 ± 0.29 | 2.18 ± 0.30 | −2.77 ± 0.31 | −0.59 ± 0.09 | 2.03 ± 0.22 | −2.78 ± 0.32 | −0.75 ± 0.23 | −0.04 | −0.10 | 0.39 | 2540 | 0.28 | 1.8 | 0.01 | −0.09 | 732 ± 43 |
| 2017 | Illecillewaet | 1.47 ± 0.19 | 1.54 ± 0.20 | −2.55 ± 0.27 | −1.01 ± 0.18 | 2.00 ± 0.16 | −2.84 ± 0.32 | −0.84 ± 0.28 | – | – | 0.36 | 2615 | 0.32 | 2.19 | 0.01 | 0 | 718 ± 49 |
| 2017 | Haig | 1.58 ± 0.20 | 1.65 ± 0.23 | −3.56 ± 0.31 | −1.91 ± 0.21 | 1.50 ± 0.17 | −3.43 ± 0.29 | −1.93 ± 0.24 | – | – | 0.04 | – | 0.31 | 1.62 | 0.01 | 0.04 | 885 ± 10 |
| 2017 | Conrad | 2.10 ± 0.22 | 1.91 ± 0.23 | −2.97 ± 0.26 | −1.06 ± 0.13 | 2.17 ± 0.17 | −3.12 ± 0.29 | −0.95 ± 0.24 | −0.16 | −0.16 | 0.48 | 2600 | 0.31 | 2.68 | 0 | −0.01 | 730 ± 45 |
| 2017 | Kokanee | 3.15 ± 0.32 | 2.86 ± 0.33 | −3.14 ± 0.34 | −0.28 ± 0.08 | 2.84 ± 0.25 | −2.87 ± 0.34 | −0.03 ± 0.23 | 0.00 | 0.01 | 0.62 | 2560 | 0.34 | 1.99 | −0.08 | −0.01 | 711 ± 55 |
| 2016 | Zillmer | 1.68 ± 0.19 | 1.72 ± 0.20 | −2.27 ± 0.22 | −0.55 ± 0.07 | 1.99 ± 0.23 | −2.61 ± 0.33 | −0.62 ± 0.24 | 0.02 | −0.38 | 0.49 | 2410 | 0.21 | 1.76 | 0.01 | −0.02 | 726 ± 46 |
| 2016 | Nordic | 1.79 ± 0.22 | 1.70 ± 0.23 | −1.85 ± 0.24 | −0.15 ± 0.08 | 1.79 ± 0.14 | −1.90 ± 0.21 | −0.11 ± 0.16 | −0.08 | 0.01 | 0.43 | 2555 | 0.16 | 1.63 | 0 | −0.04 | 727 ± 40 |
| 2016 | Illecillewaet | 1.41 ± 0.17 | 1.46 ± 0.18 | −1.73 ± 0.18 | −0.27 ± 0.05 | 1.34 ± 0.17 | −2.49 ± 0.29 | −1.15 ± 0.24 | – | – | 0.60 | 2550 | 0.45 | 1.9 | −0.01 | 0.05 | 718 ± 54 |
| 2016 | Haig | 1.15 ± 0.15 | 1.21 ± 0.17 | −2.27 ± 0.20 | −1.06 ± 0.11 | 1.88 ± 0.12 | −2.08 ± 0.20 | −0.20 ± 0.16 | 0.11 | −0.13 | 0.03 | na | 0.38 | 1.24 | −0.01 | −0.04 | 893 ± 10 |
| 2016 | Conrad | 1.40 ± 0.18 | 1.47 ± 0.19 | −1.74 ± 0.20 | −0.27 ± 0.06 | 2.07 ± 0.13 | −1.94 ± 0.26 | +0.13 ± 0.22 | −0.05 | 0.12 | 0.55 | 2530 | 0.14 | 2.1 | 0 | −0.02 | 734 ± 50 |
| 2016 | Kokanee | 1.98 ± 0.22 | 2.05 ± 0.23 | −1.93 ± 0.23 | +0.12 ± 0.05 | 2.06 ± 0.30 | −2.82 ± 0.40 | −0.76 ± 0.27 | 0.00 | −0.32 | 0.72 | 2545 | 0.15 | 1.67 | 0 | 0 | 681 ± 64 |
| 2015 | Zillmer | – | – | – | – | – | – | – | – | – | 0.30 | 2500 | – | – | – | – | – |
| 2015 | Nordic | 1.74 ± 0.22 | 1.81 ± 0.23 | −2.81 ± 0.28 | −1.0 ± 0.16 | 1.83 ± 0.19 | −3.02 ± 0.31 | −1.19 ± 0.24 | −0.16 | 0.06 | 0.32 | 2610 | 0.26 | 1.76 | 0 | 0.02 | 744 ± 42 |
| 2015 | Illecillewaet | – | – | – | – | – | – | – | – | – | 0.30 | 2600 | – | – | – | – | – |
| 2015 | Haig | – | – | – | – | 1.23 ± 0.25 | −3.02 ± 0.25 | −1.79 ± 0.25 | – | – | 0.00 | na | – | – | – | – | – |
| 2015 | Conrad | 1.65 ± 0.17 | 1.64 ± 0.18 | −3.06 ± 0.24 | −1.42 ± 0.16 | 1.80 ± 0.13 | −3.20 ± 0.35 | −1.40 ± 0.32 | −0.02 | −0.31 | 0.44 | 2685 | 0.21 | 2.2 | −0.01 | −0.03 | 736 ± 43 |
| 2015 | Kokanee | – | – | – | – | 2.18 ± 0.29 | −3.38 ± 0.40 | −1.20 ± 0.28 | 0.00 | – | 0.20 | 2680 | – | – | – | – | – |
| All | Average | 1.87 ± 0.11 | 1.88 ± 0.09 | −2.59 ± 0.16 | −0.76 ± 0.16 | 1.95 ± 0.08 | −2.67 ± 0.13 | −0.73 ± 0.15 | −0.04 | −0.14 | 0.38 | 2553 | 0.29 | 1.89 | −0.01 | −0.01 | 748 ± 62 |

**Table 4.** Density values used for geodetic and glaciological balance. Glaciological values are average values.

| Density | Geodetic $(\mathrm{kg\,m^{-3}})$ | Glaciological $(\mathrm{kg\,m^{-3}})$ | $n$ |
|---|---|---|---|
| $\rho_{\mathrm{spring}}$ | $470 \pm 70^{*}$ | $457 \pm 50^{*}$ | 74 |
| $\rho_{\mathrm{snow}}$ | $590 \pm 90$ | $570 \pm 20$ | 27 |
| $\rho_{\mathrm{firn}}$ | $700 \pm 100$ | $703 \pm 65$ | 4 |
| $\rho_{\mathrm{ice}}$ | $910 \pm 10$ | – | – |

* Geodetic spring snow density ($\rho_{\mathrm{spring}}$) is $440 \pm 50\,\mathrm{kg\,m^{-3}}$ for Haig Glacier and glaciological is $420 \pm 45\,\mathrm{kg\,m^{-3}}$ ($n = 46$).

winter and summer balance:

$$B_{\mathrm{a}} = B_{\mathrm{w}} + B_{\mathrm{s}}. \tag{1}$$

For geodetic and glaciological mass balance, we measure winter and annual balance and calculate summer balance as the difference between them:

$$B_{\mathrm{s}} = B_{\mathrm{a}} - B_{\mathrm{w}}. \tag{2}$$

To calculate geodetic winter balance ($B_{\mathrm{w\_geod}}$), we created a $\Delta$DEM from a given spring DEM and the previous late summer DEM and then applied spring snow density (Table 4). We did not independently estimate $B_{\mathrm{s\_geod}}$ because of the added uncertainty of partitioning elevation change due to melt or compaction of snow and firn.

### 2.2.2 Density estimates

While ALS provides an accurate estimate of snow depth with vertical uncertainties of $\pm 0.1$–$0.3\,\mathrm{m}$ (Abermann et al., 2010; Bollmann et al., 2011; Joerg et al., 2012), it provides no information regarding snow density. We use manual snow survey measurements available from the British Columbia River Forecast Center (BCRFC) (Weber and Litke, 2018) as independent data to estimate spring snow density, and we compare this with our measured glaciological snow densities. These snow surveys are conducted as part of the BC snow survey program eight times per year, with most sites located between 1000 and 2000 m a.s.l. We use these BCRFC data to evaluate whether reliable estimates of snow density can be obtained for regions where no snow observations over glaciers exist. The mean date of our spring field visits was 1 May (Table 2), so we chose 1 May snow survey data ($n = 10\,169$) to derive a relation between SWE ($\mathrm{kg\,m^{-2}}$) and snow depth (m) (Fig. 2). The linear relation (regression fit) yields a slope of $470 \pm 70\,\mathrm{kg\,m^{-3}}$ ($r^2 = 0.97$), which we use as the average 1 May snow density which we applied for our geodetic $B_{\mathrm{w}}$ calculations. For Haig Glacier, we chose only snow survey measurements from the Rocky Mountains for a linear relation yielding $440 \pm 50\,\mathrm{kg\,m^{-3}}$ ($n = 629$). The estimated uncertainty in bulk snow density ($\pm 70$ and $\pm 50\,\mathrm{kg\,m^{-3}}$) represents the standard deviation ($\sigma$) of the snow survey data.

For our glaciological density-informed $B_{\mathrm{w\_geod}}$, we use the observed glacier-wide snow density (Table S1 in the Supplement) and a linear regression of density versus day and used the slope ($3.0\,\mathrm{kg\,m^{-3}\,d^{-1}}$, $r^2 = 0.43$) and days between the survey and the observations to adjust for change in snow density (Fig. 3). The lack of an altitudinal trend in snow density observed on many glaciers (Fausto et al., 2018; McGrath et al., 2015, 2018; Sold et al., 2016) and those of this study, coupled with the absence of high-elevation snow density measurements and the annual variability of snow density evolution, required the use of a single value for spring snow density.

Regional observations of late summer snow density are consistent (Table 5), ranging from 530 to 630 $\mathrm{kg\,m^{-3}}$ for glaciers across the Pacific Northwest (Table 5). This is expected for temperate, midlatitude glaciers, where snow densities range from the critical density of about $550\,\mathrm{kg\,m^{-3}}$ (Benson, 1962; Herron and Langway, 1980) to around $600\,\mathrm{kg\,m^{-3}}$ depending upon regional climatology. Since we independently evaluate glaciological versus geodetic estimates of mass change, we compare application of our late summer glaciological snow density measurements to calculate net balance with estimates based on the average of typical observations from four regional sources ($590 \pm 60\,\mathrm{kg\,m^{-3}}$; Table 5), to test the impact of uncertainties of up to 10 % in this parameter. Firn density has not been reported for the study area, so we estimate $700 \pm 100\,\mathrm{kg\,m^{-3}}$ for multi-year firn based on observations in the Alps (Ambach et al., 1966). This value is also consistent with our firn core measurements for firn 2 or more years old (Table S2; average density of $703 \pm 65\,\mathrm{kg\,m^{-3}}$, $n = 4$). Measurements of 1-year-old firn averaged $619 \pm 47\,\mathrm{kg\,m^{-3}}$ ($n = 8$). Given the sustained mass loss of Pacific Northwest glaciers (Bolch et al., 2010; Menounos et al., 2019; Pelto, 2006), exposed firn is generally more than 1 year old, and we apply an uncertainty of 2 times the $\sigma$ of our multi-year firn core observations ($\pm 15$ %), which captures the range of observed firn densities (664–776 $\mathrm{kg\,m^{-3}}$). We use an ice density of $910 \pm 10\,\mathrm{kg\,m^{-3}}$ (Clarke et al., 2013). After performing a pixel-based surface classification for each late summer, we used these classification masks to assign a density (Table 4) to each pixel (snow/firn/ice).

### 2.2.3 Firn processes

Firn meltwater retention and densification are neglected in our study. Firn densification (Belart et al., 2017; Sold et al., 2013) can be modeled, but this approach assumes that net annual surface elevation change corresponds to the average annual accumulation layer transformed from end-of-year snow density to ice (Sold et al., 2013). Glaciers in this study have a low average accumulation ablation area ratio (AAR, 38 %, Table 3), CE3 and ice area ratios range from 38 % to 94 % (mean: 47 %). In most years, a significant amount of multi-year firn is exposed on these glaciers, similar to other glaciers

**The Cryosphere, 13, 1–19, 2019**

**Table 5.** Late summer snow density observations from regional studies. We use $570\,\mathrm{kg\,m^{-3}}$ as our density of late summer snow for geodetic mass balance but also separately calculate mass balance using the average for regional studies excluding those from glaciers in this study ($590\,\mathrm{kg\,m^{-3}}$).

| Location | Mean $\rho_{\mathrm{snow}}$ ($\mathrm{kg\,m^{-3}}$) | Range $\rho_{\mathrm{snow}}$ ($\mathrm{kg\,m^{-3}}$) | References |
| --- | --- | --- | --- |
| South Cascade Gl., WA, USA | 580 | 530–600 | Bidlake et al. (2010), Krimmel (1996) |
| Juneau Icefield, AK, USA | 560 | 540–580 | Miller and Pelto (1999), Pelto and Miller (1990) |
| Castle Creek Gl., BC, CA | 600 | – | Beedle et al. (2014) |
| North Cascades, WA, USA | 600 | 590–630 | Pelto and Riedel (2001) |
| Haig Glacier, AB, CA | 545 | 530–570 | Marshall (2012) |
| Columbia Basin, BC, CA | 570 | 535–615 | This study |

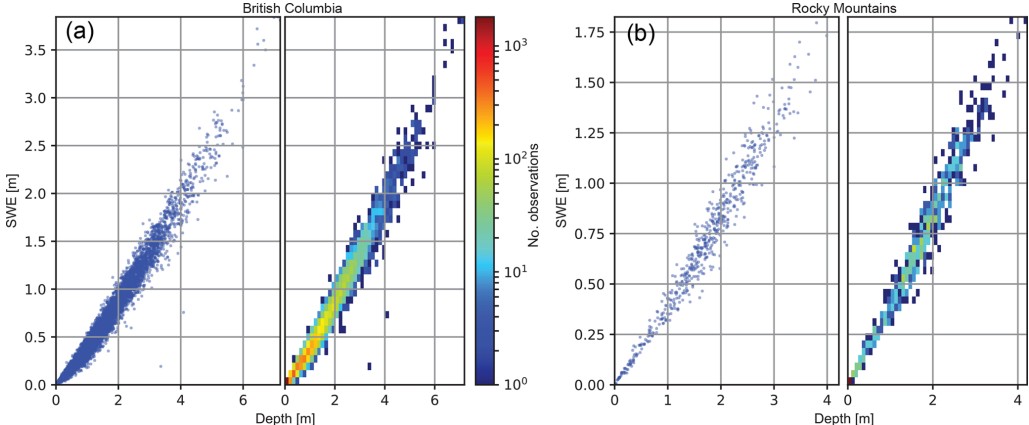

**Figure 2.** Snow depth versus snow water equivalent from 1 May provincial snow survey data. The mean date of our spring field seasons was 1 May, and so we chose 1 May BC snow survey data **(a)** to derive a SWE–snow-depth regression from which we determined the average 1 May snow density ($470 \pm 70\,\mathrm{kg\,m^{-3}}$ ($r^2 = 0.97$, $n = 10\,169$)). For Haig Glacier, we derived a regression from only snow stations within the Rocky Mountains south of Pine Pass to derive winter density ($440 \pm 50\,\mathrm{kg\,m^{-3}}$ ($r^2 = 0.97$, $n = 629$)).

experiencing strong mass loss (Fischer, 2011; Klug et al., 2018). Firn area and thickness losses interrupt the normal cycle of firn densification. Using the firn model of Sold et al. (2013) yields an estimated annual surface lowering over a given accumulation area due to densification of $\sim 0.20\,\mathrm{m}$; yet uncertainty in estimating surface lowering resulting from densification is high since we lack knowledge of the required input parameters. Because of this, and because firn densification is unlikely to produce firn densities outside the range of our estimate ($700 \pm 100\,\mathrm{kg\,m^{-3}}$), we chose not to estimate firn densification in our study. Firn compaction therefore comprises one systematic uncertainty term in our analysis.

### 2.3 Glaciological mass balance

We collected glacier mass balance measurements using the glaciological method (Cogley et al., 2011) with a two-season stratigraphic approach (Østrem and Brugman, 1991). Spring glaciological field campaigns typically occurred between mid-April and mid-May, and the summer/annual balance visits took place between mid-August and mid-September (Ta-

ble 2). Measurements of $B_{\mathrm{a}}$ and $B_{\mathrm{w}}$ allow the calculation of summer balance $B_{\mathrm{s}}$ (Eq. 1). Glacier mass balance measurements included snow depth, snow density, ablation, and kinematic GPS surveys of the glacier surface (Fig. 4).

Our methods apply to the four glaciers studied by the University of Northern British Columbia (UNBC): the Zillmer, Nordic, Conrad, and Kokanee glaciers. For Haig Glacier, winter mass balance measurements followed the same field protocols, but summer mass balance is derived from a combination of point observations and a distributed model of glacier melt (Marshall, 2014; Samimi and Marshall, 2017). The glacier melt model has 30 m resolution and uses a surface energy balance, driven by automatic-weather-station data collected on the upper glacier and in the glacier forefield. Illecillewaet Glacier has been monitored by Parks Canada since 2009 (Hirose and Marshall, 2013). We calculated $B_{\mathrm{a\_glac}}$ for Illecillewaet Glacier using the contour method since there were insufficient point measurements to estimate mass balance using the profile method.

Others have shown that snow depth is more variable than density (Elder et al., 1991; Pelto, 1996; Pulwicki et al., 2018),

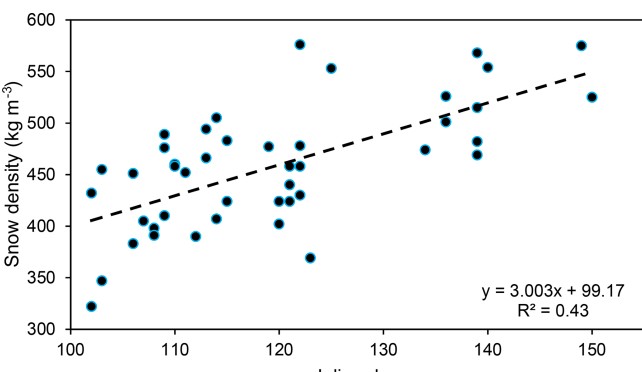

**Figure 3.** Snow density versus Julian day for all discrete snow pit and snow core locations ($n = 46$). For our glaciological density-informed estimates, we use the observed glacier-wide snow density and a linear regression of density versus day and used the slope ($3.0 \, \mathrm{kg \, m^{-3} \, d^{-1}}$ ($r^2 = 0.43$)) and days between the survey and the observations to adjust for change in snow density (Table S1).

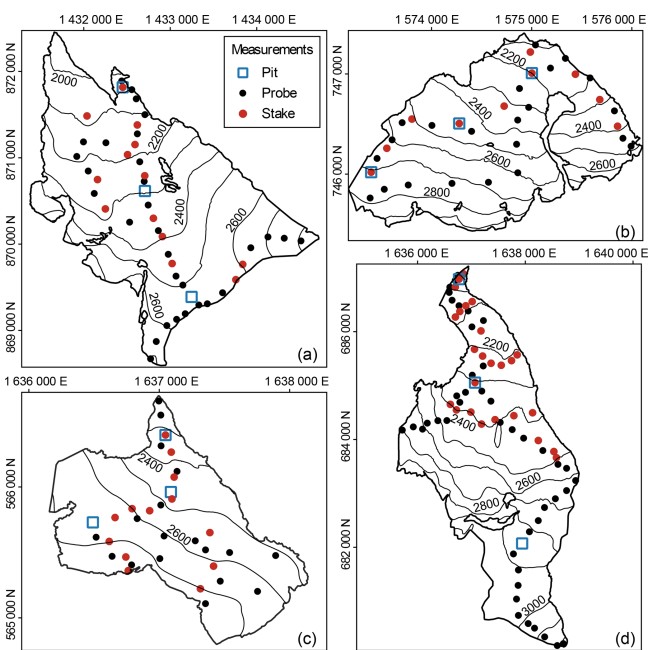

**Figure 4.** Measurement networks for the **(a)** Zillmer, **(b)** Nordic, **(c)** Kokanee, and **(d)** Conrad glaciers. Snow depth measurement locations, ablation stakes, and snow pit/core locations are pictured. Refer to Marshall (2014) for the Haig Glacier and to Hirose and Marshall (2013) for the Illecillewaet Glacier. Map coordinates are in WGS84/UTM11N.

so we designed a sampling strategy that measures snow depth much more than density (an approximate sampling ratio of 25 : 1). We used G3 industrial aluminum probes to collect over 1750 estimates of snow depth over the period of study. The probe can penetrate thick ice lenses and allowed us to measure snow depths of up to 8 m. The boundary between snow and firn is typically made up of clearly defined ice

lenses of variable thickness, which can be detected with a probe on midlatitude glaciers (Østrem and Brugman, 1991; Pelto, 1996; Sold et al., 2013). This end-of-summer surface at the glaciers in this study has such strength that an industrial probe can penetrate no more than a couple centimeters, in contrast with internal ice lenses in seasonal snowpack, which can be penetrated due to weak underlying support. Initially, we collected four probe measurements per location, but after two spring seasons we determined that two measurements were sufficient per location. The average $\sigma$ for probe measurements for four (two) measurements was 0.14 m (0.07 m) for spring and 0.10 m (0.08 m) for late summer. Two measurements per location allowed additional locations to be measured, since our observed low variability between proximal measurements is consistent with other studies (Beedle et al., 2014; Pelto et al., 2013).

We measured snow density with a $100 \, \mathrm{cm^3}$ box cutter (Hydro-Tech) in snow pits and from snow cores using a 7.25 cm diameter Kovacs corer. Our rationale to use a snow corer was that average spring snow depth exceeded 4 m and we chose to have as many sites as possible to estimate snow density. The corer also allowed us to sample internal ice lenses, which are difficult to measure with a snow sampler (Proksch et al., 2016). We measured spring snow density at low, middle, and high elevations for each glacier. If we observed an elevation trend in our density measurements, we applied a linear regression of density and elevation to our depth measurements prior to converting these data to water equivalent (mass). When there was no linear gradient, we averaged the snow density measurements to produce a glacier-wide snow density.

We conducted nine side-by-side pit–core comparisons that revealed density measured in our snow pits was comparable, with density from snow pits about $0.2 \pm 5.7 \%$ heavier than measured by subsampling snow cores (Fig. S4). The mean absolute difference between pit and core density was 4.8 %, similar to observations made at Alto dell'Ortles Glacier (Gabrielli et al., 2010). Methodological differences (Sect. S1 in the Supplement) are within the range expected between duplicate field-based measurements of snow density (1 %–6 %) and with different cutters (3 %–12 %) (Conger and McClung, 2009; Proksch et al., 2016).

Aluminum and PVC ablation stakes were used on each glacier to measure ice and firn ablation. The stake heights were measured ($\pm 1$ cm) and redrilled during each late summer visit. As a check on stake elevation, we measured depth to the previous snow surface for all stakes in firn, as stakes may self-drill in firn (Østrem and Brugman, 1991). Stakes were generally aligned along the centerline of a given glacier; however, we added a second transect of stakes to cover each branch to improve spatial coverage on each study site (Fig. 4). Conrad Glacier also featured three latitudinal sets of ablation stakes.

To calculate mass balance, we used the profile method (Escher-Vetter et al., 2009), applied over 100 m hypsometric

elevation bins. The area–altitude CE4 distribution of a given glacier was obtained using our annual late summer ALS DEMs. The boundary of each glacier was manually delineated using the ALS DEM hillshade of the previous late summer and a $\Delta$DEM (Abermann et al., 2010). We also calculated mass balance using linear regression. For Zillmer, Nordic, and Conrad glaciers, we separately considered the measurements from two distinct branches or sides of each glacier and then separately applied the profile and linear methods to each branch.

To account for mass change between a given field visit and the associated ALS survey, we completed kinematic GPS surveys using a Topcon GB-1000 receiver as a rover and a second receiver as a base station. We corrected base station data using Natural Resources Canada Precise Point Positioning (https://webapp.geod.nrcan.gc.ca/geod/tools-outils/ppp.php, last access: 1 June 2018) before post-processed surveys using Topcon Tools. Height changes observed between the ALS DEM surface and survey points were binned by elevation (Fig. S5) and assigned a density based upon surface classification as determined from satellite imagery. Since ALS surveys were essentially synchronous (typically flown over 2 to 3 d), we chose to apply the correction to the glaciological estimates of mass balance. We surveyed 2–6 control points at each site to determine the survey reliability and found that horizontal and vertical uncertainties respectively averaged $\pm 0.04$ and $\pm 0.06$ m.

## 2.4 Uncertainty assessment

We analyzed stable terrain to derive statistical indicators of bias and data dispersion from $\Delta$DEM using a late summer DEM as a reference. We report the mean, median, and normalized median absolute deviation (NMAD) over stable terrain (Table 3), which generally covered 10–20 km$^2$. To calculate uncertainty in ALS-derived height change, we also account for spatial correlation as assessed over stable terrain based on semivariogram analysis (Fig. S3) as described in Rolstad et al. (2009). We bias correct the height change over the glacier surfaces using the systematic elevation difference over stable terrain (Bias$_{\Delta h}$) in the $\Delta$DEMs (Table S3). This bias correction ranged from $-0.09$ to $0.05$ m and averaged $-0.01$ m. NMAD reveals random errors that are typically below $\pm 0.3$ m, with a maximum of $0.6$ m (Table 3). This maximum error occurred for Zillmer Glacier in late summer 2017 when the separation between site visit and ALS survey was large and new snow covered the glacier during the ALS survey (Table 2).

Random uncertainty stems from three sources that we assume to be independent: (i) elevation change uncertainty ($\sigma h_{\Delta DEM}$), (ii) glacier zone delineation uncertainty ($\sigma A$), and (iii) volume-to-mass density conversion uncertainty ($\sigma \rho$). We define elevation change uncertainty ($\sigma h_{\Delta DEM}$) following the methods detailed in Menounos et al. (2019) and found an average decorrelation length of 0.75 km (Fig. S3).

Below, we have abbreviated our geodetic and glaciological uncertainty assessment (detailed version: Sect. S2 in the Supplement).

For delineation of ice/firn/snow zones from satellite imagery (Fig. S1), we applied a buffering method (Granshaw and Fountain, 2006) to the perimeter of each zone that was not at the glacier boundary. Our satellite imagery resolution varied from 3 to 15 m, so we chose a buffer of 4 times the largest pixel size to derive an uncertainty in area per zone. This 60 m buffer accounts for uncertainty in zone delineation and changes in the positions of the zone boundaries occurring between ALS and satellite imagery acquisition dates. Total random uncertainty in volume change is

$$\sigma \Delta V = \sqrt{(\sigma h_{\Delta DEM}(p + 5(1-p))A)^2 + (\sigma A \cdot \bar{h}_{\Delta DEM})^2}, \quad (3)$$

where $A$ is the area of a given glacier and $p$ is the percentage of surveyed area, which averaged 99.1 % (Table 2). Random uncertainty on geodetic mass balance is

$$\sigma \Delta M = \sum_i \sqrt{(\sigma \Delta V_i \cdot \rho_i)^2 + (\sigma \rho_i \cdot \Delta V)^2} \cdot \frac{A_i}{A_{tot}}, \quad (4)$$

where $\rho_i$ is individual density conversion values with associated uncertainties ($\pm \sigma \rho_i$) for spring snow, late summer snow, firn, and ice (Table 4). Prior to being summed to produce a final uncertainty, each zone (ice/firn/snow) is considered separately for $B_a$, with $\Delta V_i$ and $A_i$ the volume and area change of each zone respectively.

Firn compaction or fresh snow on the surveyed surface introduce systematic uncertainty on geodetic balance. On Drangajökull ice cap, where $B_w$ is more than 1 m w.e. greater than our average $B_w$, firn compaction and fresh snow densification increased geodetic $B_w$ by 8 %. Fresh snow off-glacier was negligible in all but a few cases. We thus assume a systematic uncertainty ($\sigma \Delta M_{sys}$) of 10 % on $B_{a,w}$. Collectively, random and systematic uncertainty thus yield total uncertainty in mass balance:

$$\sigma B_{geod} = \sqrt{(\sigma \Delta M)^2 + (\sigma \Delta M_{sys})^2}. \quad (5)$$

To determine uncertainty in glaciological mass balance, we derive a mean density ($\bar{\rho}$) of mass change (Table 3) and uncertainty in height change for both observations and GPS survey corrections. Uncertainty in glaciological mass balance is calculated as

$$\sigma B_{a,w} = \sqrt{\sigma \Delta h_{glac}^2 \cdot \bar{\rho}^2 + \sigma \rho^2 \cdot B_{a,w}^2}, \quad (6)$$

where $\sigma \rho$ is the uncertainty on density taken to be 10 % of $\bar{\rho}$ to account for uncertainty in density measurements and extrapolation of those measurements. The uncertainty in extrapolation of glaciological observations to glacier-wide mass balance is taken as the $\sigma$ of the different calculations of mass balance for each season.

For both geodetic and glaciological mass balance, $B_s$ was derived as the difference of annual and summer balance (Eq. 1), and thus uncertainty on $B_s$ yields

$$\sigma B_s = \sqrt{\sigma B_a{}^2 + \sigma B_w^2}. \qquad (7)$$

## 3 Results

### 3.1 Glaciological versus geodetic balance

Comparison of seasonal balance from glaciological and geodetic methods showed strong agreement (Fig. 5), with glaciological winter balance ($B_{w\_glac}$) averaging $1.95 \pm 0.08$ m w.e., which is 4 % greater than our geodetic estimate. Average summer and annual geodetic balance estimates were 3 % more negative than our glaciological measurements (Fig. 6). CE5 For individual glaciers, average difference between $B_{a\_glac}$ and $B_{a\_geod}$ was in excellent agreement ($-0.03$ m w.e. relative to $B_{a\_glac}$), with an average absolute deviation of $0.10 \pm 0.07$ m w.e. a$^{-1}$ between estimates (Fig. 6). $B_{w\_glac}$ was 5 % greater relative to $B_{w\_geod}$, and $B_{s\_glac}$ was 4 % more positive relative to $B_{s\_geod}$ when considering individual glaciers. For $B_w$ and $B_s$, geodetic and glaciological balance were within 20 % for over 85 % of cases. Average mean annual balance from 2015 to 2017 was $-0.73 \pm 0.15$ and $-0.76 \pm 0.16$ m w.e. for glaciological and geodetic methods respectively (Table 3). Mean $B_{s\_glac}$ was $-2.67 \pm 0.13$ m w.e. All individual estimates of seasonal and annual balance are within $2\sigma$ uncertainties, and only in three instances are they outside $1\sigma$ uncertainties (Fig. 6).

We created a $\Delta$DEM from the first and last late summer DEM for each site (Fig. 7) and compared calculated mass change from this $\Delta$DEM to the sum of the individual balance years that comprised that given period (Fig. 8). We found that all cumulative seasonal $B_a$ estimates from glaciological and geodetic balance were within uncertainty ($2\sigma$) of the last–first mass change approach (Fig. 8). Glaciological balance was in net more positive (average $+0.09$ m w.e.) and had an average absolute difference of 0.20 m w.e. from the last–first $\Delta$DEM. Summed $B_{a\_geod}$ agree with our last–first estimates, with an average deviation of only 0.03 m w.e.

### 3.2 Glaciological density observations

Glacier averaged snow density from snow pits and cores for spring is $457 \pm 48$ kg m$^{-3}$, with a coefficient of variation (CV) of 0.14 ($n = 74$). This estimate is 13 kg m$^{-3}$ less than our snow-survey-based geodetic $\rho_{spring}$ but is within uncertainty (Table 4). For Haig Glacier, average spring density is $420 \pm 45$ kg m$^{-3}$, which is 20 kg m$^{-3}$ lighter than our estimate obtained from nearby snow survey measurements but again within uncertainty. Our average late summer glaciological density of $570 \pm 20$ kg m$^{-3}$ ($n = 27$) ranged from 536 to 617 kg m$^{-3}$ (CV $= 0.04$). Assigned geodetic $\rho_{snow}$ is 18 kg m$^{-3}$ greater than observations. Average probe depth

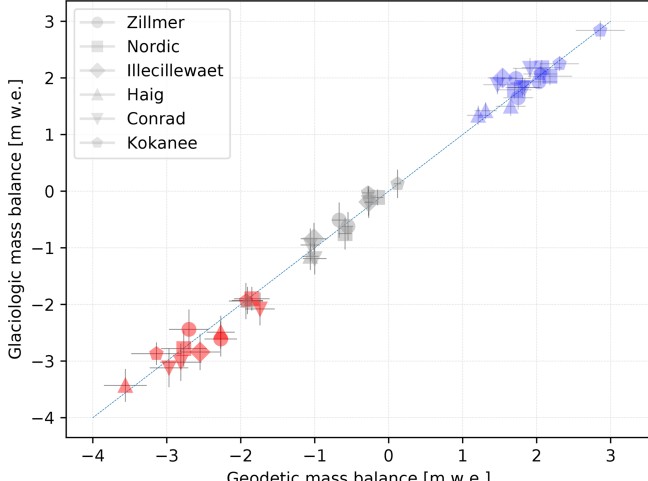

**Figure 5.** Geodetic versus glaciological mass balance estimates for 2015 through 2018 for all six study glaciers with a one-to-one line. Blue shows the winter balance covering the accumulation season from mid-September to late April; red shows the summer balance spanning the remaining months; gray shows the annual balance. Errors depicted are $1\sigma$ uncertainties. Average $B_{w\_glac}$ was 4 % greater than $B_{w\_geod}$, and $B_{s\_glac}$ and $B_{a\_glac}$ were 4 % greater than our geodetic estimates on average. CE6

for spring is $4.20 \pm 0.06$ m, with a CV of 0.33 ($n = 1754$). Average probe depth in late summer is $1.85 \pm 0.10$ m, with a CV of 0.78 ($n = 777$). Observed glacier-wide average snow depths are between 3.4 and 6.9 m and average $4.56 \pm 0.21$ m. While spring snow density showed greater variability than late summer snow density, snow depth is far more variable than snow density in both seasons.

Over the period 2015–2017 average AAR was 38 % (Table 3), with multi-year firn exposed over 13 % of the glacier surface, thus leaving the remaining 49 % of glacier area as bare ice. Located in the Rocky Mountains, Haig Glacier is the easternmost site in our study and is in a lower accumulation environment. It has lost nearly all its firn cover over the last 20 years, with firn area at 6 % in 2015. The study glaciers that lie in the Columbia Mountains had an AAR of 45 % with 15 % exposed multi-year firn cover and 40 % bare glacier ice.

#### 3.2.1 Geodetic density sensitivity

The effect of using a regional late summer snow density (Table 5) versus our glaciological density values (Table S1) depends on the amount of retained snow and glaciological density but produces a $< 0.01$ m w.e. $B_{a\_geod}$ decrease on average, which is a negligible contribution. Varying firn density by $\pm 15$ % also has an average effect on $B_{a\_geod}$ of $\pm 0.01$ m w.e., with the largest impact (0.04 m w.e.) experienced at Conrad Glacier in 2015, when 17 % of the glacier was exposed firn. However, misclassifying a given area of glacier surface has a significant impact, as $\rho_{firn}$ is 17 %

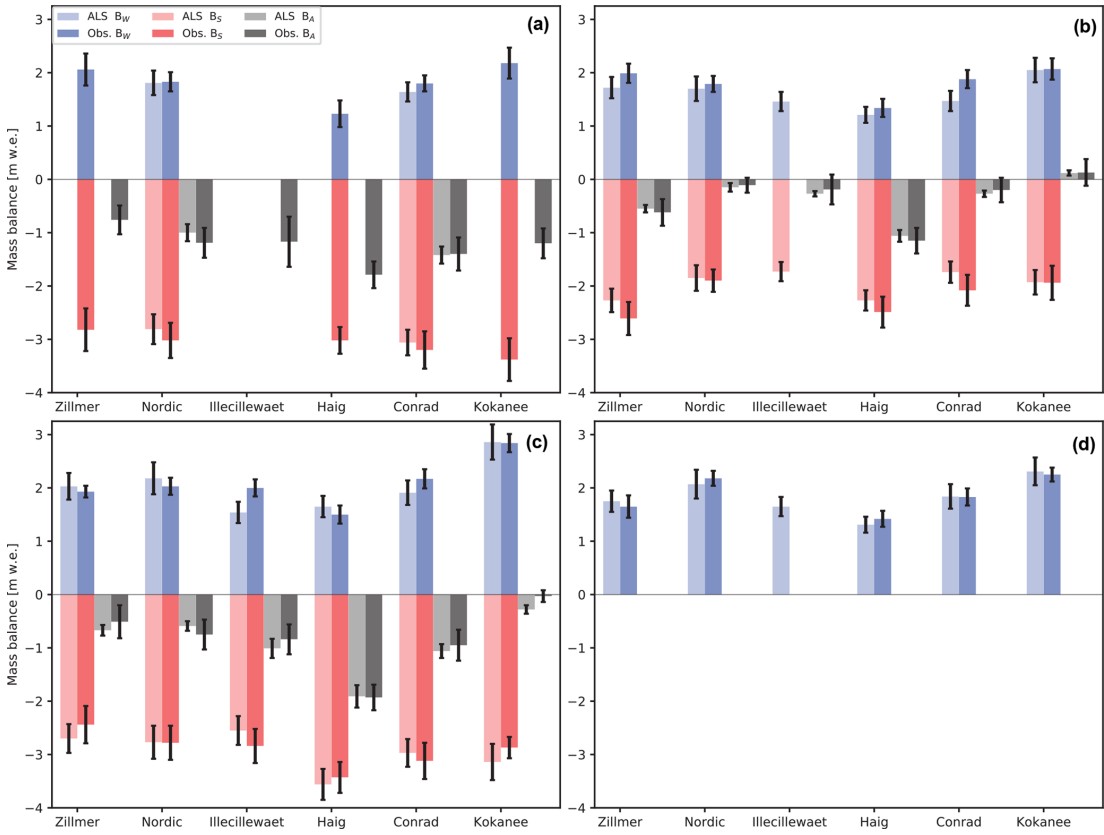

**Figure 6.** Seasonal and annual mass balance for all study glaciers from both geodetic and glaciological measurements for each balance year from 2014 to 2018 with $1\sigma$ uncertainties. **(a)** 2014 to 2015 balance year, **(b)** 2015 to 2016 balance year, **(c)** 2016 to 2017 balance year, and **(d)** 2017 to 2018 winter balance.

greater than snow and 26 % less than $\rho_{ice}$. If we produce a single glacier-wide density ($\bar{\rho}$) instead of distributing density based on surface classification, we change absolute magnitudes of $B_{a\_geod}$ by an average of $\pm 0.10$ m w.e. Though we did not use it for mass conversion, our $\bar{\rho}$ of $B_{a\_geod}$ ranged from 681 (Kokanee 2016) to 895 kg m$^{-3}$ (Haig 2017) and averaged $748 \pm 61$ kg m$^{-3}$.

Applying snow survey density values for spring snow (Table 4) versus our glaciological snow density observations (Table S1) reduces average $B_{w\_geod}$ by 0.03 m w.e. (1.7 %) and causes $B_{w\_geod}$ to be 7 % greater rather than 5 % relative to $B_{w\_glac}$. For individual glaciers, $B_{w\_geod}$ values between the two methods differ by 1 to 13 % but only 2 % on average.

## 3.3 Glaciological and geodetic balance discrepancies

Estimates of seasonal and annual balance for individual glaciers were outside $1\sigma$ uncertainties in a few cases. Conrad $B_{w\_glac}$ was 24 % greater than $B_{w\_geod}$ in 2016. Snow accumulation may have occurred in the 8 d between the Conrad Glacier ALS survey and field visit, as we observed over 1 m of fresh snow over 4 d during that interval while on Kokanee Glacier. Automatic snow weather stations near both glaciers at around 2050 m a.s.l. showed no accumulation, highlight-

ing the steep balance gradient of the Columbia Mountains. Additionally, ALS acquisition failed over the terminus of the Conrad and Illecillewaet glaciers in late summer 2015 (Table 2), and our extrapolation based upon the typical gradient over the terminus may have underestimated melt (Fig. 7). Kokanee Glacier $B_{a\_glac}$ in 2017 was 0.25 m w.e. more positive than $B_{a\_geod}$, likely due to the burial of a few ablation stakes and subfreezing temperatures which limited our ability to take adequate snow measurements. Illecillewaet Glacier $B_{w\_glac}$ in 2017 was 46 % higher than $B_{w\_geod}$, but this difference may stem from limited $B_{w\_glac}$ observations that year ($n = 3$).

## 3.4 Interannual and spatial variability

Varied climatological conditions provided a range of balance outcomes for the period of study. The lowest $B_{w\_glac}$ of the four studied winters ($1.81 \pm 0.12$ m w.e.) occurred in 2016 yet also the least mass loss with an average $B_{a\_glac}$ of $-0.36 \pm 0.17$ m w.e. (Fig. 5). The 2016–2017 winter brought the greatest snowpack of our study period, $2.08 \pm 0.18$ m w.e., yet substantial mass loss was observed (average $B_{a\_glac}$: $-0.84 \pm 0.23$ m w.e.). The balance year of 2014–2015 saw high sus-

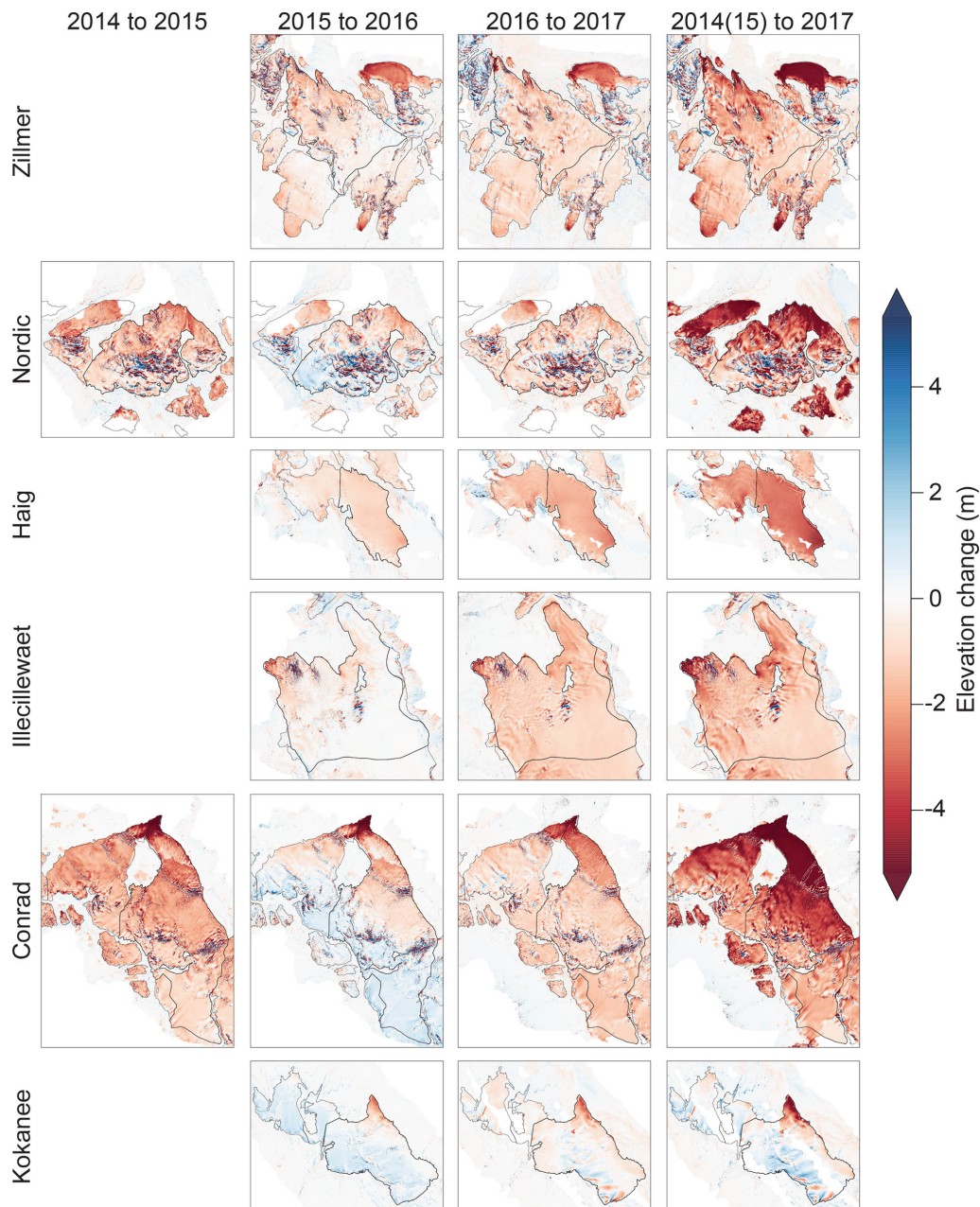

**Figure 7.** Surface height change for the Zillmer, Nordic, Haig, Illecillewaet, Conrad, and Kokanee glaciers from the first late summer DEM (2014 or 2015) until late summer 2017. Study glaciers are outlined with thick black line and other glaciers with a thin black line. Off-ice areas deemed stable terrain were used for error analysis and coregistration.

tained mass loss (average $B_{a\_glac}$ of $-1.30 \pm 0.13$ m w.e.), despite having an $B_{w\_glac}$ within 0.01 m w.e. of 2016.

The standard deviation between the seasonal and annual balances for each glacier reveals that $B_w$ ($\sigma = 0.14$ m w.e., 7 %) experiences lower interannual variability than $B_s$ ($\sigma = 0.38$ m w.e., 14 %) and $B_a$ ($\sigma = 0.35$ m w.e., 56 %). Kokanee Glacier experienced the highest $B_w$ in all four years 2015–2018 averaging $2.34 \pm 0.30$ m w.e. (Fig. 6). Haig Glacier had

the lowest $B_w$, averaging $1.37 \pm 0.11$ m w.e., and the highest mass loss (average $B_{a\_glac}$: $-1.62 \pm 0.34$ m w.e.).

We did not investigate the influence of crevasses for each glacier and each season, but for a test case for each glacier ($n = 6$) we created DEMs with filled crevasses in the late summer and then produced a $\Delta$DEM. We found that crevasse-free $\Delta$DEM $B_w$ was on average < 1 % smaller than our standard $B_w$, with discrepancies up to $-0.05$ m w.e or $-3$ %. The amount of crevassing is important, however, as

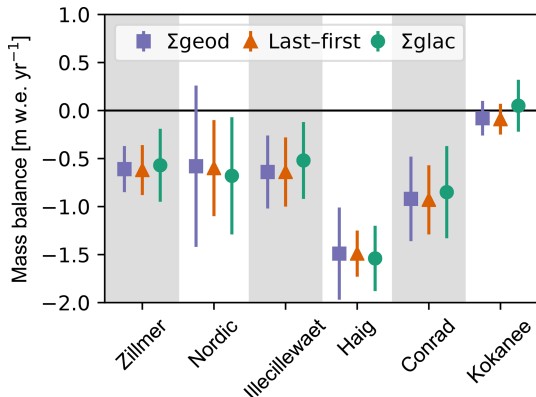

**Figure 8.** Summed annual mass balance from glaciological data ($\sum$glac), geodetic data ($\sum$geod), and last–first $\Delta$DEM. Last–first $\Delta$DEMs were created by differencing the first available DEM (2014 or 2015 late summer) from the last available DEM (2017) for each site (Table 2). Errors denote $2\sigma$ uncertainties.

some of the studied glaciers such as the Zillmer, Nordic, and Conrad feature large crevasse fields.

## 4 Discussion

The consistency between our geodetic and glaciological seasonal balance estimates among six glaciers over multiple years implies that high-resolution geodetic surveys can be used to reliably measure both winter and summer mass balance. Our study builds upon previous work that established the feasibility of geodetic methods to accurately produce $B_{\mathrm{w}}$ (Belart et al., 2017; Sold et al., 2013) and $B_{\mathrm{a}}$ (Klug et al., 2018). While others show that geodetic surveys can be applied for a single winter (Belart et al., 2017; Sold et al., 2013) or for one glacier over a number of years (Klug et al., 2018), our study demonstrates remotely measured seasonal balance is possible for widely varying rates of accumulation and ablation for multiple glaciers across entire mountain ranges.

### 4.1 Geodetic seasonal balance

Our small estimate of $\sigma h_{\Delta\mathrm{DEM}}$ (Table S3) and bias correction (Table 3) highlight that height change uncertainty is generally minor but it remains important to quantify (Joerg et al., 2012; Klug et al., 2018). As described below, density distribution and conversion factors comprise a large portion of total mass change uncertainty, with firn compaction, fresh snow at the time of ALS acquisition, and crevasses also contributing.

The spatial coverage of ALS is superior to glaciological observations; however, isolating the mass change component of surface height change at a given location is difficult and requires detailed input data (Belart et al., 2017; Sold et al., 2013). While we can develop balance gradients from glaciological data, we have not attempted to do so using our ALS

data. To date, studies have differenced their glaciological and geodetic data regarding surface height change and assigned the difference as a combination of vertical ice velocity and firn compaction (Beedle et al., 2014; Belart et al., 2017; Sold et al., 2013) or used the full-Stokes ice flow model with a bedrock DEM, a surface DEM, and in situ GPS velocities as inputs (Belart et al., 2017). Then, after applying a simple firn model, vertical ice velocity is estimated. While this method appears robust, and differencing of our glaciological observations of height change from our $\Delta$DEMs produces realistic emergence/submergence velocities, it is beyond the scope of this study.

### 4.1.1 Density distribution and conversion factors

Converting volume to mass change is a major challenge for geodetic studies (Huss, 2013; Moholdt et al., 2010). Over multiple years to decades, a constant value of density can produce tolerable uncertainty of mass change (Huss, 2013). For shorter timescales, and particularly for seasonal balance, a careful consideration of density is necessary (Klug et al., 2018). Klug et al. (2018) used ALS intensity data and satellite imagery for a pixel-based classification of the glacier surface as firn and ice. Our study built on this work and mapped areas of ice but also distinguished between snow and firn. To investigate the influence of density assumptions, we compare using independent estimates of density and our glaciological data to inform our geodetic estimates, to better constrain the uncertainty on, and compare against, glaciological seasonal balance`CE7`. Varying the density assigned to each surface class to the maximum and minimum values within our conservative uncertainties has a minor effect on seasonal balance, but failing to distribute them appropriately has a large impact. If a single density value is used, the range of values of $\bar{\rho}$ of $B_{\mathrm{a\_geod}}$ indicates that $750 \pm 60\,\mathrm{kg\,m^{-3}}$ would be most appropriate for seasonal mass change over this period (Table 3). Given the spread of $\bar{\rho}$ between glaciers, however, a glacier-specific $\bar{\rho}$ would be best.

Like Klug et al. (2018), our applied firn density was selected based on a core from a temperate glacier in the Alps (Ambach et al., 1966), and our in situ density measurements for firn $\geq 2$ years old matched this value (Table 4). Our glaciological density values for 1-year-old firn and late summer snow density are respectively 13.1 % and 22.4 % (Table 4) less than the assumed value of $700\,\mathrm{kg\,m^{-3}}$ for both snow and firn taken by Klug et al. (2018). Had we also combined snow and firn density, we would have biased $B_{\mathrm{a\_geod}}$ by varying magnitudes depending upon the surface cover. As glacier mass loss rates continue to accelerate (Menounos et al., 2019), it is reasonable to expect more and older exposed firn during the ablation season, which for geodetic studies may imply a higher density conversion factor for firn.

Applying glaciological late summer snow density versus our independent regional average density (Table 5) had little effect on $B_{\mathrm{a\_geod}}$. Future geodetic studies should use the best

available local data, however, as different regions and mountain ranges have different late summer densities (Table 5).

Using our glaciological winter density values to produce $B_{\text{w\_geod}}$ estimates resulted in a slightly greater discrepancy relative to $B_{\text{w\_glac}}$ than applying our snow-survey-based densities (Table 3). The two $B_{\text{w\_geod}}$ estimates produced similar results in net and only a 2 % average difference between $B_{\text{w\_geod}}$ estimates for individual glaciers. In the Columbia and Rocky Mountains, the first significant warming event of the spring typically occurs between early April and early May (Marshall, 2012). Springtime warming tends to homogenize and increase snow density (Adams, 1976; Elder et al., 1991). Our linear regression approach (Fig. 3) to adjust glaciological observations of spring snow density (Table S1), appears suitable over the period from mid-April through late-May, but we caution against its use for other periods of the year when densification is far slower and less predictable. For Haig Glacier, a linear relation also exists between mid-April through late May (Marshall, 2012, p. 18, Fig. 2.3). The tendency for a more homogenous snow density and lack of a consistent altitudinal trend both lend credence to using a single snow density (Fausto et al., 2018; McGrath et al., 2018).

### 4.1.2 Firn and internal processes

While firn compaction is only incorporated in our uncertainty analysis; others estimate its effect to derive $B_{\text{w\_geod}}$ (Belart et al., 2017; Sold et al., 2013) but not $B_{\text{a\_geod}}$ (Klug et al., 2018). For $B_{\text{w\_geod}}$, firn compaction was estimated based upon the annual balance in the accumulation zone over a decade (Sold et al., 2013) or over a single balance year (Belart et al., 2017). Currently accumulation areas on alpine glaciers are in constant flux and are typically discontinuous. Exposed firn is common (Fig. S1), implying that the firn zone on our study sites is shrinking in area and thickness, interrupting the cycle of firnification, and invalidating firnification models which assume that one annual layer is transformed from snow to ice annually. Nevertheless, a carefully considered treatment of firn could improve seasonal geodetic balance estimates, but as demonstrated by Belart et al. (2017), firn and fresh snow densification have little effect on $B_{\text{w\_geod}}$ if the magnitude of winter accumulation is large. For regions with low winter balance, or a colder climate, compaction would have a larger relative influence on $B_{\text{w}}$.

Meltwater retention is not incorporated into our annual balance estimates. At Haig Glacier, firn meltwater retention has not been measured, but meltwater retention in the supraglacial snowpack is a negligible contributor to mass balance, though it does create an effective energy sink, which should be accounted for in mass balance modeling (Samimi and Marshall, 2017). For glaciers in Svalbard, coupled energy balance and snow/hydrology models have been used to quantify the effects of meltwater freezing and retention on glacier mass balance (Van Pelt et al., 2012; Van Pelt and Kohler, 2015). Rates of meltwater retention are decreasing for Svalbard glaciers (Van Pelt and Kohler, 2015) and on the Devon Ice Cap (Bezeau et al., 2013), due to decreasing firn area and, in particular, warmer temperature. Like at our glaciers, melt–freeze cycles form thick summer surface layers on these Svalbard glaciers and Devon Ice Cap, which could act as a barrier for vertical water transport and is likely to promote near-surface lateral water flow, limiting deep firn water storage (Gascon et al., 2013; Van Pelt and Kohler, 2015).

Geodetic balance implicitly includes internal and basal mass change, which are not captured by the glaciological method. Studies of these processes are rare and are based upon estimates rather than verified measurements. Estimates of annual mass loss from geothermal heat, potential energy released by runoff or ice motion, and basal friction are typically around 0.01 to 0.10 m w.e. (Huss et al., 2009; Oerlemans, 2013; Sold et al., 2016). Crevasses and internal processes likely combine to be 0 % to 4 % of the magnitude of average annual ablation (e.g., Klug et al., 2018; Sold et al., 2016) and thus are likely minor contributors to seasonal balance in the Columbia Mountains. Modeled meltwater accumulation in firn would tend to increase mass balance, possibly offsetting typical basal/internal mass loss, but would not be captured by geodetic or glaciological measurements. Most mass balance models only assume vertical percolation of meltwater, yet given thick impermeable ice layers observed in our cores and snow pits, and in other studies (Gascon et al., 2013; Van Pelt and Kohler, 2015), this assumption would lead to an overestimation of refreezing. Without regional data to constrain firn processes it is difficult to incorporate them into mass balance calculations. Regionally, a better understanding of firn processes could improve annual balance and runoff estimates and likely has a greater influence on the large ice fields in western North America, which have received little attention. Although firn processes are not resolved, our approach markedly improves the quality of annual results compared to calculations based on a fixed glacier-wide conversion density.

### 4.1.3 Fresh snow

Presence of fresh snow at the time of acquisition is a challenge for any geodetic survey estimating mass change (Belart et al., 2017; Joerg et al., 2012; Klug et al., 2018). Fresh snow can change the height of the target surface by tens of centimeters. Our bias correction of $\Delta$DEM height change (Fig. S2, Table 3) corrected for small quantities of fresh snow, assuming that snow was present over stable terrain. In late summer, we could detect fresh snow visually, as a hillshade of the glacier surface at 1 m resolution captures intricate details which are easily disguised by snow depths of 0.2 m or more. Off-glacier, the depth and distribution of fresh snow is variable due to redistribution and the thermal properties of bedrock and other surfaces. In spring, we are unable to detect fresh snow as the only snow-free pixels in our scenes are

typically rock faces with extreme slopes and tree tops. Our $\sigma \Delta M_{\text{sys}}$ attempts to approximate the systematic uncertainty introduced by fresh snow and firn compaction.

### 4.1.4 Crevasses

Crevasses can affect both $B_{\text{w\_geod}}$ and $B_{\text{w\_glac}}$ since crevasses bridged by winter snowpack will overestimate $B_{\text{w\_geod}}$ snow volume, and crevasses filled by snow would not be captured by $B_{\text{w\_glac}}$. We produced crevasse-free glacier surfaces by resampling late summer DEMs to 10 m using the maximum elevations within the smoothing window to avoid in-crevasse height measurements. Using the 10 m crevasse-free DEMs versus the original 1 m DEMs had little influence on $B_{\text{w\_geod}}$, with only the Zillmer and Nordic glaciers showing a difference $> 1\%$. We did not extend these test cases to cover $B_{\text{a\_geod}}$ estimates because the area of exposed crevasses varied little year to year. On Hintereisferner, crevasse effects biased $B_{\text{a\_geod}}$ by only 0.03 % ($-0.047$ m w.e.) over a decade (Klug et al., 2018). Despite the small influence of crevassing on $B_{\text{a\_geod}}$ observed in this study, additional studies should quantify the magnitude of this bias in greater detail than presented here.

### 4.2 Glaciological seasonal balance

Observational biases include the representativeness of sampling sites and number of measurements (Cogley, 1999; Fountain and Vecchia, 1999), as well as the extrapolation of those measurements to produce a glacier-wide balance estimate (Sold et al., 2016; Thibert and Vincent, 2009). The difficulty of comparability between methods and sites (Cogley, 1999; Fountain and Vecchia, 1999) is an ongoing challenge due to logistical and financial obstacles to in situ mass balance studies. Areas of a glacier may be inaccessible, and preferred paths chosen for measurement may be biased to areas which better retain snowpack for safety purposes (Østrem and Brugman, 1991).

### 4.2.1 Snow depth

We observed best agreement between geodetic and glaciological measurements of winter balance during years of dense field surveys. Safety or logistical constraints prevented us from completing all transects of snow depth measurements in some years, with greater discrepancies between estimates in cases with incomplete coverage. In both spring and late summer, we encountered internal ice layers at some or all sites. Ice lenses were most common in the accumulation zone, but they were also found in the ablation zone in spring. These internal layers form via rain-on-snow events (McCabe et al., 2007) or, as the melt season progresses, via internal storage of meltwater (Pfeffer and Humphrey, 1996). Ice layers 2–6 cm thick were present nearly every year in the accumulation zone of the Conrad Glacier and often at other sites. We were able to penetrate these layers and successfully mea-

sure spring snow depth using our industrial avalanche probe. A conventional avalanche probe is unsuitable for glaciological observations in the Columbia Mountains.

The greater $B_{\text{w\_geod}}$ of 2016 on Conrad Glacier is likely due to both snow accumulation between the glaciological visit and ALS survey, as well as due to the late summer 2015 ALS survey missing the lowest reaches of the glacier, preventing calculation of surface height change for that portion of the glacier. We estimated the snow depth for the lower reaches of the glacier based upon the ratio of snow depth observed there for other years relative to the rest of glacier. The $B_{\text{w}}$ discrepancy for Zillmer Glacier in 2016 is likely due to glaciological sampling bias, as the east transect (Fig. 4), which has a lesser snowpack, was not sampled, and the 30 d difference between field and ALS survey date (Table 2) may not be adequately resolved by the GPS survey correction.

### 4.2.2 Mass change between measurements

Previous studies account for mass change that occurs between measurements by using a distributed temperature index model (Sold et al., 2013) or degree-day model (Belart et al., 2017), but these models do not account for snow gain. We utilized in situ GPS surveys of the glacier height which were then compared with ALS DEMs. CE8 We binned and averaged our height change estimates by 100 m elevation bands (Fig. S5) and then applied a density to each band based on satellite observations of a given surface class. Limitations in our approach include (1) fresh snowfall between the GPS and ALS surveys and (2) significant densification of the snowpack in spring. Terrain presents a further challenge to kinematic GPS survey observations. The GPS antenna is securely mounted in the backpack of a field member, but the measured height of the antenna above the glacier surface may vary due to the uneven glacier terrain, particularly during travel on steep slopes (Beedle et al., 2014).

Our median dates of late summer glaciological visits and geodetic surveys are 6 and 18 September respectively (Table 2). Snowfall can occur at any time of the year in the Columbia and Rocky Mountains (Schnorbus et al., 2014), and in late August, throughout September, and even into early October, either melt or accumulation can prevail (Marshall, 2014). Lowering of the surface via ablation post ALS survey dates (Table 2) is not accounted for and would cause an underestimated winter snowpack. While our methods are comparable year to year, and between sites, our $B_{\text{w}}$ and $B_{\text{s}}$ values are not the total amount of snow and runoff during a year. We do not include snow which falls between May and August and melts off and cannot measure ablation after our ALS survey or glaciological visit, whichever occurs later. Our $B_{\text{w}}$ and $B_{\text{s}}$ values thus represent a conservative estimate of runoff contributions from snow and ice melt.

## 5  Conclusions

Estimates of seasonal mass balance presented here show strong agreement between glaciological and geodetic methods for individual glaciers and are within $1\sigma$ uncertainties for average winter, summer, and annual balance. These independent estimates of seasonal mass change accord over 3 years from glaciers separated by hundreds of kilometers. Our findings suggest that high-resolution geodetic methods, such as from ALS (Klug et al., 2018; Sold et al., 2013), aerial photogrammetry (Nolan et al., 2015), and stereo satellite imagery (Belart et al., 2017; Berthier et al., 2014), can be used to produce accurate seasonal and annual balance estimates over large areas. The quality of geodetic annual balance estimates depends more on distributing density via surface classification (Klug et al., 2018) than on the density values themselves. The spatial coverage, density of observations, and measurement precision of high-resolution geodetic terrain analysis compensate for uncertainty associated with fresh snow and firn compaction, internal and basal mass change, and crevasses (Belart et al., 2017; Klug et al., 2018). The minimal impact of these factors on mass balance stems from the large mass changes observed at our sites, as reported elsewhere (Belart et al., 2017; Klug et al., 2018). For glaciers with low mass turnover, errors introduced by firn compaction, crevasses, and fresh snow may be considerably larger than observed in our study.

Our estimate of spring snow density for geodetic measurements from provincial snow survey observations (Fig. 2) is within the uncertainty of our measured glaciological spring snow density (Table 4). Our approach holds promise for being able to use regional density estimates when in situ measurements are unavailable, yet discrepancies of up to 13 % between geodetic and glaciological winter balance estimates indicate the uncertainty introduced when using density values which are not site-specific. Estimates of end-of-season snow density introduce a possible bias, but given the regional consistency of late summer snow density and the overall lack of a density–altitude gradient in spring, using a single snow density is a robust method for converting snow depth to water equivalence (Fausto et al., 2018; McGrath et al., 2018). We observed greater variability in $B_s$ relative to $B_w$, highlighting the need for models of glacier mass balance that can be able to reliably reproduce widely varying rates of mass change corresponding to the multitude of energy fluxes that influence alpine glaciers (Fitzpatrick et al., 2017).

The hydrologic cycle of western North America is dominated by snowfall in the mountains, but observations of alpine snowpack above 2000 m a.s.l. are sparse. As the climate continues to change, there is a growing need for a more detailed understanding of the seasonal balance of glaciers and snowpack. Geodetic methods are needed to supplement in situ observations across many mountain regions in order to address the contribution of glaciers to changes in freshwater runoff availability and to sea level rise. To date, the majority of high-resolution geodetic balance studies of seasonal or annual balance have been conducted in the European Alps, where extensive, multi-decadal glaciological data are available (Klug et al., 2018; Sold et al., 2013, 2016). Our study suggests that geodetic methods can be used to assess seasonal balance of glaciers, even in mountain ranges lacking long-term records of mass balance, if density is carefully considered (Belart et al., 2017; Klug et al., 2018). Recent advances in high-resolution, optical satellite imagery (Berthier et al., 2014; Marti et al., 2016) suggest that such efforts can be made with increasing spatial and temporal coverage, greatly adding to our understanding of the seasonal contribution of snow and glaciers to mountain hydrology.

*Data availability.* Glaciological mass balance data will be available through the WGMS (https://doi.org/10.5904/wgms-fog-2018-06, WGMS, 2018).

*Supplement.* The supplement related to this article is available online at: https://doi.org/10.5194/tc-13-1-2019-supplement.

*Author contributions.* All authors contributed to the writing of the manuscript. BMP conducted the field work with the help of many invaluable volunteers and conducted the ALS processing and analysis.

*Competing interests.* The authors declare that they have no conflict of interest.

*Acknowledgements.* The authors wish to thank Amaury Dehecq for assistance with DEM coregistration. We thank two anonymous referees and the handling editor, Chris Derksen, for providing detailed reviews that substantially improved our manuscript.

*Financial support.* This research has been supported by the Columbia Basin Trust, BC Hydro, the Pacific Institute for Climate Solutions, the Natural Resources and Engineering Research Council of Canada, the Canada Chairs Program, the Tula Foundation, and the Canada Foundation for Innovation. Funding for Ben M. Pelto was provided via the Pacific Institute for Climate Solutions, the University of Northern British Columbia, and the Columbia Basin Trust. CE9

*Review statement.* This paper was edited by Chris Derksen and reviewed by two anonymous referees.

Please note the remarks at the end of the manuscript.

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

**Remarks from the language copy-editor**

CE1    Please note the short summary has a 500 character limit. I've shortened it to fit within the character limit. Please confirm or provide an alternative.

CE2    The capitalization is correct as these are proper nouns (Columbia Mountains and Rocky Mountains).

CE3    The comma is grammatically correct as what follows is an independent clause.

CE4    I've changed this to reflect your intended meaning. Thanks for clarifying.

CE5    Please note that only minor technical and necessary changes are allowed at this stage. All other changes require editor approval. Please provide an explanation for the editor for this change.

CE6    Please note that only minor technical and necessary changes are allowed at this stage. All other changes require editor approval. Please provide an explanation for the editor for this change.

CE7    Please note that only minor technical and necessary changes are allowed at this stage. We cannot remove this sentence without approval from the editor. Please provide an explanation for the editor.

CE8    Please note that only minor technical and necessary changes are allowed at this stage. All other changes require editor approval. Please provide an explanation for the editor for the changes to this paragraph.

CE9    Please verify the new financial support section and the changes to the acknowledgements.

**Remarks from the typesetter**

TS1    Please provide date of last access.

TS2    Please provide date of last access.