# Peer review of "Multi-year Evaluation of Airborne Geodetic Surveys to Estimate Seasonal Mass Balance, Columbia and Rocky Mountains, Canada"

_The Cryosphere, 2019_

## Referee Comment (RC1) · Anonymous Referee #1 · 25 Mar 2019

General comments This paper uses data collected from repeat airborne laser altimetry surveys to determine seasonal mass balance of 6 glaciers in the Columbia and Rocky mountains. Results indicate strong agreement between the geodetic estimations of mass balance and in situ observations. Technically, the paper is quite strong as it includes validation with in situ measurement and a rigorous analysis of the geodetic measurement error, however I found in many cases the writing was unclear (see comments below). In encourage to authors to carefully re-read the manuscript in order to address the many grammatical errors, improper/over use of hyphens and custom terminology throughout. Results from this work however do contribute scientifically to methods and knowledge pertaining to mass balance of Canada's western glaciers of

which are an important but diminishing source of freshwater at the head of several major watersheds. I recommend this paper for publication after addressing the comments below.

Specific comments • The meaning of 'Glacier-wise' is unclear, but unfortunately used quite oftern throughout the paper. Please change to a more intuitive term.

• Use 'our glaciers' should be changed to something less possessive like, 'glaciers in this study'.

• 279: Assuming that the exposed old firn occurs in the ablation zone, can you please provide an explanation as to how the overlying snow/firn/ice has ablated away without filling up the available pore space of the 'old firn' and leading to more internal accumulation than is accounted for in this study? this needs to be addressed as it also applies to your discussion on internal accumulation (L415-419) where it is similarly dismissed as insignificant.

• Introduction doesn't justify this work well enough. Need to elaborate on the recent trends experienced by glaciers in western Canada as per menounous et al, 2018., and the potential impacts of declining contributions to stream flow post ∼2040ish as per Clarke et al, 2015. Contributions from glacier melt to sea level rise are of secondary importance from this region as it is poorly quantified as to how much actually makes it to tidewater and how much is taken up enroute through groundwater storage and human usage.

• L28-29 re: 'Measurement of seasonal mass change provides...' - I assume your talking about in situ mass balance measurements? if so, then should be specific about it - seasonal balances can be derived from more than just in situ meausrmeents –as you indicate below.

• L37-41: poorly written paragraph.

• L50-51. "The climate of the Columbia Mountains is transitional between maritime

and continental (Demarchi, 2011), with a strong maritime influence (Hägeli and Mc-Clung, 2003)." - So its more maritime then than continental? Would inner montane better describe the climate type here?

âĞ́ L55-56: Please give average snowfall rates and specify the source and what elevations they were measured at. This is probably the aspect of the climate that is the most important for this study. âĞ́ L64-65: please quantify differences - ie. average temp, snow precip, total precip, etc. describing the differences between climate regimes as "colder and drier.." is not very informative.

âĞ́ L68-71: please link glaciers to the Columbia and Rocky mountain ranges (described above) more clearly. An outline or some indication of the extent of each of the major mountain ranges in Figure 1 would be useful

âĞ́ L77: indicate swath widths for each instrument/altitude.

âĞ́ L81: is there a systematic bias in error of the laser shots as a function of off-nadir angle? Ie., does accuracy of z degrade towards the swath margins?

âĞ́ L119: It would be helpful to add a sentence or 2 here to describe what 'snow course' data is.

âĞ́ L282-283: ' Excluding this site, the remaining study glaciers in the Columbia Mountains had an AAR of 0.45 with 0.15 exposed multi-year firn cover and 0.40 bare glacier ice.' - The way this is written it implies Haig is in the Columbia mtns, it is not.

âĞ́ L282: I presume you mean the average AAR of the remaining glaciers in the Columbia mtns? If so, please edit.

âĞ́ Line 279-283: Line 279-280 indicates firn/glacier ice extents as percentages (13% and 49%) while the same are expressed as ratios on 281 – 282. Need to be consistent. âĞ́ L373: 'In western Canada, onset of snow melt is occurring earlier on average relative to 1970-2006'. Please clarify for what period the onset of earlier snow melt is occurring.

• L387-389: the statement 'We also chose not to apply a firn correction since it requires glaciological measurements that we purposely withheld in order to evaluate the feasibility of measuring seasonal balance without surface observations from the glaciers.' Is vague. Please be more specific.

• L407-409: Re: 'Our field operations have been impacted by the melting out of crevasses: as strongly negative years are becoming the norm, and glacier flux is likely decreasing, crevasses are exposed for longer periods of time, and slower to close.

• L408: ' Please define the 'melting out of crevasses'.

• L408: What are you basing the assumption that flux is decreasing? Decreasing velocity or surface mass balance? Or both? If these assumptions are based on velocity changes, please indicate the sources used.

• L409-411: re: 'This means that the total void area of crevasses is increasing due to ablation, which we have observed on Conrad, Zillmer, Nordic, and Haig glaciers, which could possibly increase their influence on Bw.'

o Can you expand on how this was observed? Was it measured? If so, how was it measured and over what period of time?

• L415-419: Methods to measure internal accumulation include repeat shallow ice cores and ground penetrating radar (Bezeau et al., 2013; Gascon et al., 2013). As the issue of internal accumulation has not properly been addressed in western Canada, particularly over the larger icefields where this process has potential to be significant, it is worth highlighting as an important knowledge gap concerning glacier mass balance in this region.

• L426-427: re: 'Our glaciological measurement densities ranged from 0.5 to 18.5 points km-2 (Table2), whereas our ALS data had around one million points km-2.' This is an unfair concluding statement as the datasets have different limitations that are not fully discussed.

• L433-434 specify, ' as the melt season progresses. . .' ice layers may form as internal storage 'within the snowpack'

Technical corrections • Questionable use of hyphens throughout the paper.

• L281: lower accumulation, no hyphen.

• L282: 0.06 add ' km2'

• Figures: text is of variable font and size – this should be standardized for all figures. Text is so small it is unreadable on figures 7 and 4

References: Bezeau, P, Sharp M, Burgess D, and Gascon G (2013) Firn profile changes in response to extreme 21st century melting at Devon Ice Cap, Nunavut, Canada, J. Glaciol., 59(217), 981-991 (doi:10.3189/2013JoG12J208).

Gascon G, Sharp, M, Burgess D, Bezeau P, and Bush ABG (2013) Changes in accumulation-area firn stratigraphy and meltwater flow during a period of climate warming: Devon Ice Cap, Nunavut, Canada. J. Geophys. Res.-Earth, 118, 2380-2391

<hr>

---

## Referee Comment (RC2) · Anonymous Referee #2 · 26 Mar 2019

Pelto and colleagues present a detailed evaluation of airborne geodetic datasets, obtained between 2014 and 2018, to estimate seasonal mass balances on 6 glaciers in the Columbia and Rocky Mountains. This study puts airborne laser scanner (ALS) dataset with exceptional spatial and temporal resolution at its full value. The comparison of these geodetic results with the glaciological balances from in-situ measurements are carried out in a very thorough way, including an error assessment according to best international practises. I found this paper to be well thought out and mostly well described. However, in some cases the writing was a bit unclear (see comments below). The results from this work do contribute scientifically to methods and knowledge pertaining to mass balance and I would recommend the paper to be considered for

publication after consideration of the following points and a list of suggestions for minor corrections and clarifications:

DTM-related uncertainties:

Which interpolation algorithms have been used for DTM production? Please state in one paragraph!

The authors use the mean, median and normalized median absolute deviation (NMAD) of the DTM-differencing over selected stable terrain as "systematic" uncertainty (Bias$\Delta$h). It is unclear through the manuscript, how overall hdDEM was determined? Please describe in more detail! Furthermore, is hdDEM the same as Bias$\Delta$h? If so, please streamline through the manuscript!

The authors have stated not to use their gridded data for correction of their sample sizes for spatial autocorrelation (L103-104). Please state what was used instead (point cloud data?) and, if so, have the point cloud datasets also been co-registered beforehand? To calculate uncertainty in ALS-derived height change, the authors account for spatial correlation, assessed over stable terrain based on semi-variogram analysis as described in Rolstad et al. (2009). How did the authors determine $\sigma$h and what are the values of $\sigma$hdDEM Please describe in more detail and present the values in a Table (also possible in Supplement).

Specific comments

L19: delete aiborne , redundant to ALS

L26ff: I recommend to align the references based on the date of appearance. Streamline through the manuscript.

L28: "'Measurement of seasonal mass change provides..." , you mean in situ measurements or do you refer to all methods?

L35: Please rephrase sentence!

L37-41: This paragraph could be improved, giving a bit more substance!

L51: abbreviation CBR has to be introduced earlier

L68-71: Please indicate the extent of the major mountain ranges in Figure 1.

L69: try to omit redundant information, which improves readability: eg. rephrase sentence to (1) Zillmer Glacier (5.4 km2) in the Cariboo Mountains, (2) Nordic Glacier (3.4 km2) and (3) Illecillewaet Glacier (7.7 km2) in the Selkirk Mountains, (4) Conrad Glacier (11.5 km2) and (5) Kokanee Glacier (1.8 km2) in the Purcell Mountains, and (6) Haig Glacier (2.6 km2), which straddles the continental divide in the Rocky Mountains.

L77: swatch change to swath

L82: Please state in one paragraph, which interpolation algorithms you used!

L117: It would be helpful to describe what 'snow course' data is.

L256: 'glacier-wise' is an unclear term, but used quite often throughout the manuscript. Is it possible to change to another more intuitive term?

L281: omit hyphen in lower-accumulation

L282: Which unit does 0.06 have? Think km$^2$? I thought Haig glacier is not in the Columbia Mountains?

L331: $\sigma$hdDEM is not in Table 3, see comment above

L373: 'In western Canada, onset of snow melt is occurring earlier on average relative to 1970-2006'. For what period is the onset of earlier snow melt occurring? Please give detrails.

L407-411: 'Our field operations have been impacted by the melting out of crevasses: as strongly negative years are becoming the norm, and glacier flux is likely decreasing, crevasses are exposed for longer periods of time, and slower to close. This means that the total void area of crevasses is increasing due to ablation, which we have observed on Conrad, Zillmer, Nordic, and Haig glaciers, which could possibly increase their influence on Bw.'

This part is a bit unclear! What is meant with melting out of crevasses, please clarify. Which flux decreases (surface velocity or mass balance or both)? Give references of the source your assumption is based on! How was the increase of the void area of crevasses due to ablation observed? Can you detail this?

L426: The statement "Our glaciological measurement densities ranged from 0.5 to 18.5 points km-2 (Table2), whereas our ALS data had around one million points km-2" is a bit of comparing pears with apples. Please discuss in more detail or omit!

Figures: text is of variable font and size within figures. Especially on figures 4 and 7 the text is hardly readable

---

## Author Comment (AC2) · 14 May 2019

Author Response to RC2

Anonymous Referee #2

We thank the referee for their valuable comments and time spent on evaluating our manuscript.

RC2: DTM-related uncertainties:

RC2: Which interpolation algorithms have been used for DTM production? Please state in one paragraph!

[Figure]

Laser survey point density has been added to Table 2 for all ALS surveys. We added the following lines: "We post-processed point clouds and exported finished LAS files into LAStools (https://rapidlasso.com/lastools/) from which we used las2DEM to create 1 m resolution DEMs. Las2dem triangulates ground classified ALS points from las/laz files into a temporary triangulated irregular network (TIN). A DEM is then created from this using nearest neighbor interpolation. Given an average point density of greater than 2 points m-2 (Table 2), little interpolation was required." L111-115.

RC2: The authors use the mean, median and normalized median absolute deviation (NMAD) of the DTM-differencing over selected stable terrain as "systematic" uncertainty (Bias$h$). It is unclear through the manuscript, how overall hdDEM was determined? Please describe in more detail! Furthermore, is hdDEM the same as Bias$h$? If so, please streamline through the manuscript!

hdDEM is the same as Bias$\Delta h$ and should not have been included. We have removed all references to hdDEM. Bias$\Delta h$ is "...the mean height difference over stable terrain between two DEMs after coregistration". L129.

Elevation change uncertainty ($\sigma h \Delta DEM$) was calculated as a $2\sigma$ uncertainty using Supplemental eqns. 1 and 2: "Elevation change uncertainty is derived from the $\sigma$ of height change over stable terrain ($\sigma h$) after correction for effective sample size (Neff)". Lines 29-35 in S2.

Values of $\sigma h \Delta DEM$ for both Ba and Bw can now be found in Table S3.

RC2: The authors have stated not to use their gridded data for correction of their sample sizes for spatial autocorrelation (L103-104). Please state what was used instead (point cloud data?) and, if so, have the point cloud datasets also been co-registered beforehand? To calculate uncertainty in ALS-derived height change, the authors account for spatial correlation, assessed over stable terrain based on semi-variogram analysis as described in Rolstad et al. (2009). How did the authors determine h and what are the values of hdDEM Please describe in more detail and present the values in a Table

(also possible in Supplement).

Our wording was not clear. We use our gridded data to calculate effective sample size while accounting for spatial autocorrelation. The lines detailing this are in the supplemental information (S2), lines 28-37:

"Stable terrain generally covered 10-20 km2. We determined L by plotting semivariance (Figure S3) for randomly selected coordinate pairs (n=10,000) against distance for ten separate simulations and defined L as the distance at which semivariance becomes asymptotic (5% change threshold). Decorrelation length averaged 0.75 km and varied from 0.5 to 1.3 km".

We have also further clarified our treatment of spatial autocorrelation directly in the manuscript: "To calculate uncertainty in ALS-derived height change, we also account for spatial correlation as assessed over stable terrain based on semivariogram analysis (Figure S3) as described in Rolstad et al. (2009)." L246-247.

Values of $\sigma h\Delta DEM$ for both Ba and Bw can now be found in Table S3.

RC2: Specific comments

RC2: L19: delete aiborne, redundant to ALS

Removed

RC2: L26ff: I recommend to align the references based on the date of appearance. Streamline through the manuscript.

The standard for The Cryosphere (Copernicus Publications) is alphabetical, as reflected throughout the manuscript.

RC2: L28: "'Measurement of seasonal mass change provides: : :'", you mean in situ measurements or do you refer to all methods?

This was also pointed out by referee #1. We refer to both methods as either produces

results relevant to assessing meteorological drivers of glacier nourishment and melt. Revised:

RC2: "Measurement of seasonal mass change via in situ and geodetic methods provides a means to assess the importance of meteorological drivers of glacier nourishment and melt". L36-38.

L35: Please rephrase sentence!

Corrected.

RC2: L37-41: This paragraph could be improved, giving a bit more substance!

Similar to the concerns raised by Referee #1, we substantially revised the introduction of our paper.

RC2: L51: abbreviation CBR has to be introduced earlier

This acronym is now omitted since we only used it once in the paper.

RC2: L68-71: Please indicate the extent of the major mountain ranges in Figure 1.

The major mountain ranges, the Columbia and Rocky Mountains are now labeled in Figure 1 and described in the figure caption: "The Columbia and Rocky mountains are separated by the Rocky Mountain Trench (RMT).

RC2: L69: try to omit redundant information, which improves readability: eg. rephrase sentence to (1) Zillmer Glacier (5.4 km2) in the Cariboo Mountains, (2) Nordic Glacier (3.4 km2) and (3) Illecillewaet Glacier (7.7 km2) in the Selkirk Mountains, (4) Conrad Glacier (11.5 km2) and (5) Kokanee Glacier (1.8 km2) in the Purcell Mountains, and (6) Haig Glacier (2.6 km2), which straddles the continental divide in the Rocky Mountains.

Excellent suggestion. We have reorganized as suggested.

RC2: L77: swatch change to swath

Corrected.

RC2: L82: Please state in one paragraph, which interpolation algorithms you used!

Addressed in an earlier comment.

RC2: L117: It would be helpful to describe what 'snow course' data is.

We have changed 'snow course' to 'snow survey' throughout the document, as this is the official name of the BC snow survey program. We have also added a reference (Weber and Litke, 2018) that details the methodology for the BC snow survey program. The data can be found at: https://catalogue.data.gov.bc.ca/dataset/705df46f-e9d6-4124-bc4a-66f54c07b228. We now introduce the snow surveys as 'manual snow survey measurements' and have added further description of these surveys:

"These snow surveys are conducted as part of the BC snow survey program eight times per year, with most sites located between 1000 and 2000 m asl". L147-149.

RC2: L256: 'glacier-wise' is an unclear term, but used quite often throughout the manuscript. Is it possible to change to another more intuitive term?

Good suggestion, this term is unclear. We have removed all instances of 'glacier-wise'. See line 302 for an example of a revision:

"Glacier averaged snow density from snow pits and cores for spring is 457 $\pm$ 48 kg m-3. . .".

RC2: L281: omit hyphen in lower-accumulation

Removed.

RC2: L282: Which unit does 0.06 have? Think km2? I thought Haig glacier is not in the Columbia Mountains?

The value 0.06 was unit-less as it is the ratio of firn area to total glacier area, but we now use 6% for clarity, and have switched all ratios in the paragraph to percent for consistency.

RC2: We have added the following sentence to clarify the glacier locations relative to the Columbia and Rocky Mountains:

"Located in the Rocky Mountains, Haig Glacier is the easternmost site in our study and is in a lower accumulation environment. It has lost nearly all its firn cover over the last 20 years, with firn area at 6% in 2015. The study glaciers that lie in the Columbia Mountains had an AAR of 45% with 15% exposed multi-year firn cover and 40% bare glacier ice." L312-315.

RC2: L331: hdDEM is not in Table 3, see comment above

hdDEM is the same as Bias$\triangle$h and has thus been removed from the manuscript. Bias$\triangle$h is in Table 3, and we have added Table S3 which contains elevation change uncertainty ($\sigma$h$\triangle$DEM), calculated as a $2\sigma$ uncertainty using Supplemental eqns. 1 and 2.

RC2: L373: 'In western Canada, onset of snow melt is occurring earlier on average relative to 1970-2006'. For what period is the onset of earlier snow melt occurring? Please give detrails.

We have removed this sentence from the manuscript.

RC2: L407-411: 'Our field operations have been impacted by the melting out of crevasses: as strongly negative years are becoming the norm, and glacier flux is likely decreasing, crevasses are exposed for longer periods of time, and slower to close. This means that the total void area of crevasses is increasing due to ablation, which we have observed on Conrad, Zillmer, Nordic, and Haig glaciers, which could possibly increase their influence on Bw.' This part is a bit unclear! What is meant with melting out of crevasses, please clarify. Which flux decreases (surface velocity or mass balance or both)? Give references of the source your assumption is based on! How was the increase of the void area of crevasses due to ablation observed? Can you detail this?

These lines have been removed from the manuscript. The authors feel that these lines add confusion and are a distraction. We have now added to the sentence leading into these lines which now reads:

"Despite the small influence of crevassing on Ba_geod observed in this study, additional studies should quantify the magnitude of this bias in greater detail". L457-458.

What we intended to convey was that our visual field observations indicate that crevasses are being exposed (snow cover melted off) for a greater duration of the melt season than previously experienced. This extended exposure, tends to melt the sidewalls of the crevasses, widening the crevasses. After several years or decades of increased melt, many crevasses are merging to form ice-falls or serac fields that are difficult or impossible to navigate. This has implications for the safety and feasibility of travel during field work, but also for geodetic studies, as this likely increases the void area of crevasse fields, if not crevasse field extent. Ablation within crevasses is typically not captured by field studies, and may not be adequately captured in geodetic studies, depending on resolution and other factors.

As the length of the above explanation demonstrates, including these lines is a distraction from the goals of the manuscript, and while of scientific interest, our study has not taken steps to quantify these observations. Our primary goal was to highlight an area of uncertainty that future studies should tackle in greater detail, which the revised line above now does, without introducing a speculative discussion that we can add little to.

RC2: L426: The statement "Our glaciological measurement densities ranged from 0.5 to 18.5 points km-2 (Table 2), whereas our ALS data had around one million points km-2" is a bit of comparing pears with apples. Please discuss in more detail or omit!

Complete agreement here. This statement is unfair and has been removed. To discuss the relative strengths and weaknesses of each method, which this statement fails to do, is not the purpose of this section.

RC2: Figures: text is of variable font and size within figures. Especially on figures 4 and 7 the text is hardly readable

We have standardized our figure text font and size, and now text in Figures 4 and 7 is legible.

---

## Author Response (AR1)

*Author response to Anonymous Referee #1*

**We thank the referee for providing valuable feedback on our manuscript.**

Specific comments The meaning of 'Glacier-wise' is unclear, but unfortunately used quite oftern throughout the paper. Please change to a more intuitive term.

**Good suggestion, this term is unclear. We have removed all instances of 'glacier-wise'. See line 302 for an example of a revision:**

*"Glacier averaged snow density from snow pits and cores for spring is 457 ± 48 kg m-3…".*

Use 'our glaciers' should be changed to something less possessive like, 'glaciers in this study'.

**We agree with this suggestion and have changed instances of 'our study glaciers' and 'our glaciers' to more appropriate phrasing such as: 'glaciers in this study', 'studied glaciers', and 'these glaciers'.**

279: Assuming that the exposed old firn occurs in the ablation zone, can you please provide an explanation as to how the overlying snow/firn/ice has ablated away without filling up the available pore space of the 'old firn' and leading to more internal accumulation than is accounted for in this study? this needs to be addressed as it also applies to your discussion on internal accumulation (L415-419) where it is similarly dismissed as insignificant.

**This is a fair criticism. We now state in section 2.2.3:**

*"Firn meltwater retention and densification are neglected in our study."*

**We also added a discussion of meltwater retention renaming our section 4.1.2 Firn compaction, to 'Firn and internal processes'**

*"Internal accumulation within firn is not not incorporated into our annual balance estimates as it is not component of surface mass balance and is not measured within geodetic or glaciological surface mass balance studies. At Haig Glacier, firn meltwater retention has not been measured, but meltwater retention in the supraglacial snowpack is a negligible contributor to mass balance, though it does create an effective "energy sink", that should be accounted for in mass balance modeling (Samimi and Marshall, 2017). For glaciers in Svalbard, coupled energy balance and snow/hydrology models have been used to quantify the effects of meltwater freezing and retention on glacier mass balance (Van Pelt et al., 2012; Van Pelt and Kohler, 2015). Rates of meltwater retention are decreasing for Svalbard glaciers (Van Pelt and Kohler, 2015), and on the Devon Ice Cap (Bezeau et al., 2013), due to decreasing firn area and in particular, warmer temperature. Like at our glaciers, melt-freeze cycles form thick 'summer surface' layers on these Svalbard glaciers and Devon Ice Cap, which could act as a barrier for vertical water transport and is likely to promote near-surface lateral water flow, limiting deep firn water storage (Gascon et al., 2013; Van Pelt and Kohler, 2015)." L417-426.*

Introduction doesn't justify this work well enough. Need to elaborate on the recent trends experienced by glaciers in western Canada as per menounous et al, 2018., and the potential impacts of declining

contributions to stream flow post 2040ish as per Clarke et al, 2015. Contributions from glacier melt to sea level rise are of secondary importance from this region as it is poorly quantified as to how much actually makes it to tidewater and how much is taken up enroute through groundwater storage and human usage.

**We agree with the reviewer that the importance of glacier mass change is more about water resources and much less about sea level rise. We revised the introduction to emphasize the importance of mass change on water resources.**

L28-29 re: 'Measurement of seasonal mass change provides...' - I assume your talking about in situ mass balance measurements? if so, then should be specific about it - seasonal balances can be derived from more than just in situ meausrmeents –as you indicate below.

**A fair point. Sentence is now modified to,** *"Measurement of seasonal mass change via in situ and geodetic methods provides a means to assess the importance of meteorological drivers of glacier nourishment and melt". L36-38.*

L37-41: poorly written paragraph.

**We refined this poorly written paragraph.**

L50-51. "The climate of the Columbia Mountains is transitional between maritime and continental (Demarchi, 2011), with a strong maritime influence (Hägeli and Mc-Clung, 2003)." - So its more maritime then than continental? Would inner montane better describe the climate type here?

**We have changed the sentence to:** *"The climate of the Columbia Mountains is transitional between maritime and continental (Demarchi, 2011). L72.*

L55-56: Please give average snowfall rates and specify the source and what elevations they were measured at. This is probably the aspect of the climate that is the most important for this study.

**An excellent suggestion.  Please see the next comment response for average winter precipitation.**

L64-65: please quantify differences - ie. average temp, snow precip, total precip, etc. describing the differences between climate regimes as "colder and drier.." is not very informative.

**We now refer to climate normals from 1981-2010 for two nearby weather stations in the Columbia and Rocky Mountains. The text now states:**

*"From 1981-2010, Rodgers Pass, located in the center of the Columbia Mountains (Figure 1), at an elevation of 1330 m, has an average annual temperature of +1.9˚C, and an average winter (December-February) temperature of -8.0˚C, and experiences 1056 ± 49 mm w.e. of precipitation through the accumulation season (October-April) (Environment Canada, 2019)". L77-80.*

*"From 1981-2010, Lake Louise, located in the center of the southern Canadian Rockies (Figure 1), at an elevation of 1524 m, had an average annual temperature of +0.2˚C, an average winter temperature of -11.6˚C, and experienced 298 ± 9 mm w.e. of precipitation through the accumulation season. As*

*evidenced by comparing Lake Louise and Rodgers Pass, the Canadian Rockies are drier and colder in winter than the Columbia Mountains." L88-92.*

**While these two stations alone are not representative of their entire respective mountain ranges, they do quantitatively demonstrate the metrological differences between the two climate regimes.**

L68-71: please link glaciers to the Columbia and Rocky mountain ranges (described above) more clearly. An outline or some indication of the extent of each of the major mountain ranges in Figure 1 would be useful

**We have added the following sentence to clarify the glacier locations relative to the Columbia and Rocky Mountains:**

*"Haig Glacier is in the Rocky Mountains, whereas the other five glaciers lie in the Columbia Mountains". L99-100.*

**The major mountain ranges, the Columbia and Rocky Mountains are now labeled in Figure 1 and described in the figure caption:**

*"The Columbia and Rocky Mountains are separated by the Rocky Mountain Trench (RMT)".*

L77: indicate swath widths for each instrument/altitude.

**We added swath widths for each instrument:**

*"The VQ-580 and Q-780 were respectively flown at flying heights of around 500 and 2500 m above the terrain that yielded swath widths of 500-1000 m and 2000-3000 m". L105-106.*

L81: is there a systematic bias in error of the laser shots as a function of off-nadir angle? Ie., does accuracy of z degrade towards the swath margins?

**A good point raised by the referee. Yes, off-nadir laser shots do have larger positional errors than nadir ones. We designed flight lines to have 53% overlap, and this overlap would tend to reduce off-nadir bias. Any bias introduced by this sampling should be captured in the height uncertainty that we calculate for the stable terrain. We have expanded lines 106-108 to clarify this:**

*"We planned laser surveys with 53% overlap between flight lines, to yield return point densities that averaged 1-3 laser shots m$^{-2}$ (Table 2) with an effective sampling diameter of 10-20 cm per laser shot.". Point density has been added to Table 2 for all laser surveys.*

L119: It would be helpful to add a sentence or 2 here to describe what 'snow course' data is.

**We have changed 'snow course' to 'snow survey' throughout the document, as this is the official name of the BC snow survey program. We have also added a reference (Weber and Litke, 2018) that details the methodology for the BC snow survey program. The data can be found at: https://catalogue.data.gov.bc.ca/dataset/705df46f-e9d6-4124-bc4a-66f54c07b228. We now introduce the snow surveys as 'manual snow survey measurements' and have added further description of these surveys:**

*"These snow surveys are conducted as part of the BC snow survey program eight times per year, with most sites located between 1000 and 2000 m asl". L147-149.*

L282-283: 'Excluding this site, the remaining study glaciers in the Columbia Mountains had an AAR of 0.45 with 0.15 exposed multi-year firn cover and 0.40 bare glacier ice.' - The way this is written it implies Haig is in the Columbia mtns, it is not.

**Sentence is now modified to:**
**"The study glaciers that lie in the Columbia Mountains had an AAR of 45% with 15% exposed multi-year firn cover and 40% bare glacier ice." L314-315.**

L282: I presume you mean the average AAR of the remaining glaciers in the Columbia mtns? If so, please edit.

**See the above comment for the clarified sentence.**

Line 279-283: Line 279-280 indicates firn/glacier ice extents as percentages (13% and 49%) while the same are expressed as ratios on 281 – 282. Need to be consistent.

**We have switched all ratios in the paragraph to percent for consistency.**

L373: 'In western Canada, onset of snow melt is occurring earlier on average relative to 1970-2006'. Please clarify for what period the onset of earlier snow melt is occurring.

**We have removed this sentence from the manuscript.**

L387-389: the statement 'We also chose not to apply a firn correction since it requires glaciological measurements that we purposely withheld in order to evaluate the feasibility of measuring seasonal balance without surface observations from the glaciers.' Is vague. Please be more specific.

**This statement has been removed from the manuscript. We initially chose to produce geodetic winter balance estimates only using the snow survey density to evaluate the feasibility of measuring seasonal balance without surface observations from the glaciers. However, we then used our in-situ densities to produce a separate geodetic winter balance estimate for each glacier to assess the impact of using in-situ versus regional density values (Table 3). The statement was attempting to convey that firnification models require an estimate of accumulation zone balance, which geodetic measurements, without correcting for ice dynamics, cannot provide.**

L407-409: Re: 'Our field operations have been impacted by the melting out of crevasses: as strongly negative years are becoming the norm, and glacier flux is likely decreasing, crevasses are exposed for longer periods of time, and slower to close.
L408: ' Please define the 'melting out of crevasses'.
L408: What are you basing the assumption that flux is decreasing? Decreasing velocity or surface mass balance? Or both? If these assumptions are based on velocity changes, please indicate the sources used.
L409-411: re: 'This means that the total void area of crevasses is increasing due to ablation, which we have observed on Conrad, Zillmer, Nordic, and Haig glaciers, which could possibly increase their influence on Bw.' Can you expand on how this was observed? Was it measured? If so, how was it

measured and over what period of time?

**The lines referenced in the above four referee comments have been removed from the manuscript. The authors feel that these lines add confusion and are a distraction. We have now added to the sentence leading into these lines which now reads:**

***"Despite the small influence of crevassing on Ba_geod observed in this study, additional studies should quantify the magnitude of this bias in greater detail".***

**What we intended to convey was that our visual field observations indicate that crevasses are being exposed (snow cover melted off) for a greater duration of the melt season than previously experienced. This extended exposure, tends to melt the sidewalls of the crevasses, widening the crevasses. After several years or decades of increased melt, many crevasses are merging to form ice-falls or serac fields that are difficult or impossible to navigate. This has implications for the safety and feasibility of travel during field work, but also for geodetic studies, as this likely increases the void area of crevasse fields, if not crevasse field extent. Ablation within crevasses is typically not captured by field studies, and may not be adequately captured in geodetic studies, depending on resolution and other factors.**

**As the length of the above explanation demonstrates, including these lines is a distraction from the goals of the manuscript, and while of scientific interest, our study has not taken steps to quantify these observations. Our primary goal was to highlight an area of uncertainty that future studies should tackle in greater detail, which the revised line above now does, without introducing a speculative discussion that we can add little to.**

L415-419: Methods to measure internal accumulation include repeat shallow ice cores and ground penetrating radar (Bezeau et al., 2013; Gascon et al., 2013). As the issue of internal accumulation has not properly been addressed in western Canada, particularly over the larger icefields where this process has potential to be significant, it is worth highlighting as an important knowledge gap concerning glacier mass balance in this region.

**We agree with this comment. We have now highlighted this important knowledge gap and added discussion of this process in section 4.1.2 as detailed in a previous comment. See lines 417–426 for the added material.**

L426-427: re: 'Our glaciological measurement densities ranged from 0.5 to 18.5points km-2 (Table2), whereas our ALS data had around one million points km-2.' This is an unfair concluding statement as the datasets have different limitations that are not fully discussed.

**We concur with the referee here and have removed this statement from the manuscript.**

L433-434 specify, ' as the melt season progresses: : :' ice layers may form as internal storage 'within the snowpack'

**Amended as suggested.**

Technical corrections
Questionable use of hyphens throughout the paper.

**Thank you for highlighting this issue. We have double-checked (sorry for the pun) our use of hyphens and corrected as requested.**

L281: lower accumulation, no hyphen.

**Hyphen removed.**

L282: 0.06 add ' km2'

**This was a ratio, and now is expressed as a percent.**

Figures: text is of variable font and size – this should be standardized for all figures.
Text is so small it is unreadable on figures 7 and 4

**We standardized our figure text font and size, so text in Figures 4 and 7 is legible.**

References: Bezeau, P, Sharp M, Burgess D, and Gascon G (2013) Firn profile changes in response to extreme 21st century melting at Devon Ice Cap, Nunavut, Canada, J. Glaciol., 59(217), 981-991 (doi:10.3189/2013JoG12J208).

Gascon G, Sharp, M, Burgess D, Bezeau P, and Bush ABG (2013) Changes in accumulation-area firn stratigraphy and meltwater flow during a period of climate warming: Devon Ice Cap, Nunavut, Canada. J. Geophys. Res.-Earth, 118, 2380-2391

**Thank you for these references, in addition to (Van Pelt et al., 2012; Van Pelt and Kohler, 2015), these were very informative to read and better discuss firn processes.**

*Author response to Anonymous Referee #2*

**We thank the referee for their valuable comments and time spent on evaluating our manuscript.**

DTM-related uncertainties:

Which interpolation algorithms have been used for DTM production? Please state in one paragraph!

**Laser survey point density has been added to Table 2 for all ALS surveys. We added the following lines:**
**"We post-processed point clouds and exported finished LAS files into LAStools (https://rapidlasso.com/lastools/) from which we used las2DEM to create 1 m resolution DEMs. Las2dem triangulates ground classified ALS points from las/laz files into a temporary triangulated irregular network (TIN). A DEM is then created from this using nearest neighbor interpolation. Given an average point density of greater than 2 points m$^{-2}$ (Table 2), little interpolation was required."**
**L111-115.**

The authors use the mean, median and normalized median absolute deviation (NMAD) of the DTM-differencing over selected stable terrain as "systematic" uncertainty (Bias$h$). It is unclear through the manuscript, how overall hdDEM was determined? Please describe in more detail! Furthermore, is hdDEM the same as Bias$h$? If so, please streamline through the manuscript!

**$\bar{h}_{dDEM}$ is the same as Bias$_{\Delta h}$ and should not have been included. We have removed all references to $\bar{h}_{dDEM}$. Bias$_{\Delta h}$ is "...the mean height difference over stable terrain between two DEMs after coregistration". L129.**

**Elevation change uncertainty ($\sigma h_{\Delta DEM}$) was calculated as a 2$\sigma$ uncertainty using Supplemental eqns. 1 and 2: "Elevation change uncertainty is derived from the $\sigma$ of height change over stable terrain ($\sigma h$) after correction for effective sample size ($N_{eff}$)". Lines 29-35 in S2.**

**Values of $\sigma h_{\Delta DEM}$ for both Ba and Bw can now be found in Table S3.**

The authors have stated not to use their gridded data for correction of their sample sizes for spatial autocorrelation (L103-104). Please state what was used instead (point cloud data?) and, if so, have the point cloud datasets also been co-registered beforehand? To calculate uncertainty in ALS-derived height change, the authors account for spatial correlation, assessed over stable terrain based on semi-variogram analysis as described in Rolstad et al. (2009). How did the authors determine h and what are the values of hdDEM Please describe in more detail and present the values in a Table (also possible in Supplement).

**Our wording was not clear. We use our gridded data to calculate effective sample size while accounting for spatial autocorrelation. The lines detailing this are in the supplemental information (S2), lines 28-37:**

**"Stable terrain generally covered 10-20 km2. We determined L by plotting semivariance (Figure S3) for randomly selected coordinate pairs (n=10,000) against distance for ten separate simulations and defined L as the distance at which semivariance becomes asymptotic (5% change threshold). Decorrelation length averaged 0.75 km and varied from 0.5 to 1.3 km".**

**We have also further clarified our treatment of spatial autocorrelation directly in the manuscript:** *"To calculate uncertainty in ALS-derived height change, we also account for spatial correlation as assessed over stable terrain based on semivariogram analysis (Figure S3) as described in Rolstad et al. (2009)." L246-247.*

**Values of σh$_{\Delta DEM}$ for both Ba and Bw can now be found in Table S3.**

Specific comments

L19: delete aiborne, redundant to ALS

**Removed**

L26ff: I recommend to align the references based on the date of appearance. Streamline through the manuscript.

**The standard for The Cryosphere (Copernicus Publications) is alphabetical, as reflected throughout the manuscript.**

L28: "'Measurement of seasonal mass change provides: : :" , you mean in situ measurements or do you refer to all methods?

**This was also pointed out by referee #1. We refer to both methods as either produces results relevant to assessing meteorological drivers of glacier nourishment and melt. Revised:**

*"Measurement of seasonal mass change via in situ and geodetic methods provides a means to assess the importance of meteorological drivers of glacier nourishment and melt". L36-38.*

L35: Please rephrase sentence!

**Corrected.**

L37-41: This paragraph could be improved, giving a bit more substance!

**Similar to the concerns raised by Referee #1, we substantially revised the introduction of our paper.**

L51: abbreviation CBR has to be introduced earlier

**This acronym is now omitted since we only used it once in the paper.**

L68-71: Please indicate the extent of the major mountain ranges in Figure 1.

**The major mountain ranges, the Columbia and Rocky Mountains are now labeled in Figure 1 and described in the figure caption:** *"The Columbia and Rocky mountains are separated by the Rocky Mountain Trench (RMT).*

L69: try to omit redundant information, which improves readability: eg. rephrase sentence to (1) Zillmer Glacier (5.4 km2) in the Cariboo Mountains, (2) Nordic Glacier (3.4 km2) and (3) Illecillewaet Glacier (7.7 km2) in the Selkirk Mountains, (4) Conrad Glacier (11.5 km2) and (5) Kokanee Glacier (1.8 km2) in the Purcell Mountains, and (6) Haig Glacier (2.6 km2), which straddles the continental divide in the Rocky Mountains.

**Excellent suggestion. We have reorganized as suggested.**

L77: swatch change to swath

**Corrected.**

L82: Please state in one paragraph, which interpolation algorithms you used!

**Addressed in an earlier comment.**

L117: It would be helpful to describe what 'snow course' data is.

**We have changed 'snow course' to 'snow survey' throughout the document, as this is the official name of the BC snow survey program. We have also added a reference (Weber and Litke, 2018) that details the methodology for the BC snow survey program. The data can be found at: https://catalogue.data.gov.bc.ca/dataset/705df46f-e9d6-4124-bc4a-66f54c07b228. We now introduce the snow surveys as 'manual snow survey measurements' and have added further description of these surveys:**

*"These snow surveys are conducted as part of the BC snow survey program eight times per year, with most sites located between 1000 and 2000 m asl". L147-149.*

L256: 'glacier-wise' is an unclear term, but used quite often throughout the manuscript. Is it possible to change to another more intuitive term?

**Good suggestion, this term is unclear. We have removed all instances of 'glacier-wise'. See line 302 for an example of a revision:**

*"Glacier averaged snow density from snow pits and cores for spring is 457 ± 48 kg m-3…".*

L281: omit hyphen in lower-accumulation

**Removed.**

L282: Which unit does 0.06 have? Think km2? I thought Haig glacier is not in the Columbia Mountains?

**The value 0.06 was unitless as it is the ratio of firn area to total glacier area, but we now use 6% for clarity, and have switched all ratios in the paragraph to percent for consistency.**

**We have added the following sentence to clarify the glacier locations relative to the Columbia and Rocky Mountains:**

*"Located in the Rocky Mountains, Haig Glacier is the easternmost site in our study and is in a lower accumulation environment. It has lost nearly all its firn cover over the last 20 years, with firn area at 6% in 2015. The study glaciers that lie in the Columbia Mountains had an AAR of 45% with 15% exposed multi-year firn cover and 40% bare glacier ice." L312-315.*

L331: hdDEM is not in Table 3, see comment above

**$\bar{h}_{dDEM}$ is the same as Bias$_{\Delta h}$ and has thus been removed from the manuscript. Bias$_{\Delta h}$ is in Table 3, and we have added Table S3 which contains elevation change uncertainty (σhΔDEM), calculated as a 2σ uncertainty using Supplemental eqns. 1 and 2.**

L373: 'In western Canada, onset of snow melt is occurring earlier on average relative to 1970-2006'. For what period is the onset of earlier snow melt occurring? Please give detrails.

**We have removed this sentence from the manuscript.**

L407-411: 'Our field operations have been impacted by the melting out of crevasses: as strongly negative years are becoming the norm, and glacier flux is likely decreasing, crevasses are exposed for longer periods of time, and slower to close. This means that the total void area of crevasses is increasing due to ablation, which we have observed on Conrad, Zillmer, Nordic, and Haig glaciers, which could possibly increase their influence on Bw.' This part is a bit unclear! What is meant with melting out of crevasses, please clarify. Which flux decreases (surface velocity or mass balance or both)? Give references of the source your assumption is based on! How was the increase of the void area of crevasses due to ablation observed? Can you detail this?

**These lines have been removed from the manuscript. The authors feel that these lines add confusion and are a distraction. We have now added to the sentence leading into these lines which now reads:**

*"Despite the small influence of crevassing on Ba_geod observed in this study, additional studies should quantify the magnitude of this bias in greater detail". L457-458.*

**What we intended to convey was that our visual field observations indicate that crevasses are being exposed (snow cover melted off) for a greater duration of the melt season than previously experienced. This extended exposure, tends to melt the sidewalls of the crevasses, widening the crevasses. After several years or decades of increased melt, many crevasses are merging to form ice-falls or serac fields that are difficult or impossible to navigate. This has implications for the safety and feasibility of travel during field work, but also for geodetic studies, as this likely increases the void area of crevasse fields, if not crevasse field extent. Ablation within crevasses is typically not captured by field studies, and may not be adequately captured in geodetic studies, depending on resolution and other factors.**

**As the length of the above explanation demonstrates, including these lines is a distraction from the goals of the manuscript, and while of scientific interest, our study has not taken steps to quantify these observations. Our primary goal was to highlight an area of uncertainty that future studies should tackle in greater detail, which the revised line above now does, without introducing a speculative discussion that we can add little to.**

L426: The statement "Our glaciological measurement densities ranged from 0.5 to 18.5 points km-2 (Table2), whereas our ALS data had around one million points km-2" is a bit of comparing pears with apples. Please discuss in more detail or omit!

**Complete agreement here. This statement is unfair and has been removed. To discuss the relative strengths and weaknesses of each method, which this statement fails to do, is not the purpose of this section.**

Figures: text is of variable font and size within figures. Especially on figures 4 and 7 the text is hardly readable

**We have standardized our figure text font and size, and now text in Figures 4 and 7 is legible.**

[revised manuscript text omitted]
}$ ± σ$_{geod.bw}$ | Bw$_{geod.sc}$ ± σ$_{geod.bw}$ | Bs$_{geod}$ ± σ$_{geod.bs}$ | Ba$_{geod}$ ± σ$_{geod.ba}$ | Bw$_{glac}$ ± σ$_{glac.bw}$ | Bs$_{glac}$ ± σ$_{glac.bs}$ | Ba$_{glac}$ ± σ$_{glac.ba}$ | Bw$_{surv.corr}$ | Ba$_{surv.corr}$ | AAR | ELA (m) | NMAD Ba (m) | NMAD Bw (m) | Median Ba$_{\Delta h}$ (m) | Bias$_{\Delta h}$ (m) | p̄ (kg m$^{-3}$) |
|---|---|---|---|---|---|---|---|---|---|---|---|---|---|---|---|---|---|
| 2018 | Zillmer | 1.70 ± 0.19 | 1.75 ± 0.20 | | | 1.65 ± 0.17 | | | -0.15 | | | | | 1.4 | | | |
| 2018 | Nordic | 1.87 ± 0.26 | 2.07 ± 0.27 | | | 2.18 ± 0.14 | | | -0.04 | | | | | 1.76 | | | |
| 2018 | Illecillewaet | 1.61 ± 0.17 | 1.65 ± 0.18 | | | — | | | na | | | | | 2.26 | | | |
| 2018 | Haig | 1.25 ± 0.15 | 1.31 ± 0.19 | | | 1.42 ± 0.15 | | | na | | | | | 1.83 | | | |
| 2018 | Conrad | 1.62 ± 0.21 | 1.84 ± 0.23 | | | 1.83 ± 0.12 | | | 0.00 | | | | | 2.34 | | | |
| 2018 | Kokanee | 2.07 ± 0.25 | 2.31 ± 0.26 | | | 2.25 ± 0.13 | | | 0.01 | | | | | 1.76 | | | |
| 2017 | Zillmer | 2.12 ± 0.24 | 2.03 ± 0.25 | -2.70 ± 0.27 | -0.67 ± 0.10 | 1.93 ± 0.26 | -2.44 ± 0.35 | -0.51 ± 0.23 | 0.15 | -0.31 | 0.48 | 2440 | 0.6 | 1.83 | -0.1 | -0.05 | 729 ± 45 |
| 2017 | Nordic | 2.14 ± 0.29 | 2.18 ± 0.30 | -2.77 ± 0.31 | -0.59 ± 0.09 | 2.03 ± 0.22 | -2.78 ± 0.32 | -0.75 ± 0.23 | -0.04 | -0.10 | 0.39 | 2540 | 0.28 | 1.8 | 0.01 | -0.09 | 732 ± 43 |
| 2017 | Illecillewaet | 1.47 ± 0.19 | 1.54 ± 0.20 | -2.55 ± 0.27 | -1.01 ± 0.18 | 2.00 ± 0.16 | -2.84 ± 0.32 | -0.84 ± 0.28 | — | — | 0.36 | 2615 | 0.32 | 2.19 | 0.01 | 0 | 718 ± 49 |
| 2017 | Haig | 1.58 ± 0.20 | 1.65 ± 0.23 | -3.56 ± 0.31 | -1.91 ± 0.21 | 1.50 ± 0.17 | -3.43 ± 0.29 | -1.93 ± 0.24 | — | — | 0.04 | na | 0.31 | 1.62 | 0.01 | 0.04 | 885 ± 10 |
| 2017 | Conrad | 2.10 ± 0.22 | 1.91 ± 0.23 | -2.97 ± 0.26 | -1.06 ± 0.13 | 2.17 ± 0.17 | -3.12 ± 0.29 | -0.95 ± 0.24 | -0.16 | -0.16 | 0.48 | 2600 | 0.31 | 2.68 | 0 | -0.01 | 730 ± 45 |
| 2017 | Kokanee | 3.15 ± 0.32 | 2.86 ± 0.33 | -3.14 ± 0.34 | -0.28 ± 0.08 | 2.84 ± 0.25 | -2.87 ± 0.34 | -0.03 ± 0.23 | 0.00 | 0.01 | 0.62 | 2560 | 0.34 | 1.99 | -0.08 | -0.01 | 711 ± 55 |
| 2016 | Zillmer | 1.68 ± 0.19 | 1.72 ± 0.20 | -2.27 ± 0.22 | -0.55 ± 0.07 | 1.99 ± 0.23 | -2.61 ± 0.33 | -0.62 ± 0.24 | 0.02 | -0.38 | 0.49 | 2410 | 0.21 | 1.76 | 0.01 | -0.02 | 726 ± 46 |
| 2016 | Nordic | 1.79 ± 0.22 | 1.70 ± 0.23 | -1.85 ± 0.24 | -0.15 ± 0.08 | 1.79 ± 0.14 | -1.90 ± 0.21 | -0.11 ± 0.16 | -0.08 | 0.01 | 0.43 | 2555 | 0.16 | 1.63 | 0 | -0.04 | 727 ± 40 |
| 2016 | Illecillewaet | 1.41 ± 0.17 | 1.46 ± 0.18 | -1.73 ± 0.18 | -0.27 ± 0.05 | — | — | -0.19 ± 0.28 | — | — | 0.60 | 2550 | 0.45 | 1.9 | -0.01 | 0.05 | 718 ± 54 |
| 2016 | Haig | 1.15 ± 0.15 | 1.21 ± 0.17 | -2.27 ± 0.20 | -1.06 ± 0.11 | 1.34 ± 0.17 | -2.49 ± 0.29 | -1.15 ± 0.24 | — | — | 0.03 | na | 0.38 | 1.24 | -0.01 | -0.04 | 893 ± 10 |
| 2016 | Conrad | 1.40 ± 0.18 | 1.47 ± 0.19 | -1.74 ± 0.20 | -0.27 ± 0.06 | 1.88 ± 0.12 | -2.08 ± 0.20 | -0.20 ± 0.16 | 0.11 | -0.13 | 0.55 | 2530 | 0.14 | 2.1 | 0 | -0.02 | 734 ± 50 |
| 2016 | Kokanee | 1.98 ± 0.22 | 2.05 ± 0.23 | -1.93 ± 0.23 | +0.12 ± 0.05 | 2.07 ± 0.13 | -1.94 ± 0.26 | +0.13 ± 0.22 | -0.05 | 0.12 | 0.72 | 2545 | 0.15 | 1.67 | 0 | 0 | 681 ± 64 |
| 2015 | Zillmer | — | — | — | — | 2.06 ± 0.30 | -2.82 ± 0.40 | -0.76 ± 0.27 | 0.00 | -0.32 | 0.30 | 2500 | — | — | — | — | — |
| 2015 | Nordic | 1.74 ± 0.22 | 1.81 ± 0.23 | -2.81 ± 0.28 | -1.0 ± 0.16 | 1.83 ± 0.19 | -3.02 ± 0.31 | -1.19 ± 0.24 | -0.16 | 0.06 | 0.32 | 2610 | 0.26 | 1.76 | 0 | 0.02 | 744 ± 42 |
| 2015 | Illecillewaet | — | — | — | — | — | — | -1.17 ± 0.47 | — | — | 0.30 | 2600 | — | — | — | — | — |
| 2015 | Haig | — | — | — | — | 1.23 ± 0.25 | -3.02 ± 0.25 | -1.79 ± 0.25 | — | — | 0.00 | na | — | — | — | — | — |
| 2015 | Conrad | 1.65 ± 0.17 | 1.64 ± 0.18 | -3.06 ± 0.24 | -1.42 ± 0.16 | 1.80 ± 0.13 | -3.20 ± 0.35 | -1.40 ± 0.32 | -0.02 | -0.31 | 0.44 | 2685 | 0.21 | 2.2 | -0.01 | -0.03 | 736 ± 43 |
| 2015 | Kokanee | — | — | — | — | 2.18 ± 0.29 | -3.38 ± 0.40 | -1.20 ± 0.28 | 0.00 | — | 0.20 | 2680 | — | — | — | — | — |
| All | Average | 1.84 ± 0.11 | 1.88 ± 0.09 | -2.59 ± 0.16 | -0.72 ± 0.16 | 1.95 ± 0.08 | -2.71 ± 0.13 | -0.70 ± 0.15 | -0.04 | -0.14 | 0.38 | 2553 | 0.29 | 1.89 | -0.01 | -0.01 | 748 ± 62 |

[revised manuscript text omitted]

---

## Editor Decision (ED1)

[revised manuscript text omitted]